# Expansion of the geranylgeranyl pyrophosphate synthase gene family underlies the evolution of terpenoid biosynthesis in termites

Natan Horáček[1,2], Ondřej Lukšan[1], Zarley Rebholz[3], Karel Harant[1], Radek Pohl[1],
Lana Mutabdžija-Nedelcheva[1,4], Simon Hellemans[5], Daniel Jungwirth[1,6], Jan Křivánek[1],
Anna Amirianová[1,7], Pavlína Kyjaková[1], Thomas Bourguignon[5], Dorothea Tholl[8],
Robert Hanus[1]*, Jitka Štáfková [1]*

1 Institute of Organic Chemistry and Biochemistry of the Czech Academy of Sciences, Prague, Czech
Republic, 2 Department of Analytical Chemistry, Faculty of Science, Charles University, Prague, Czech
Republic, 3 Department of Biology, Indiana University, Bloomington, Indiana, United States of America,
4 Department of Genetics and Microbiology, Faculty of Science, Charles University, Prague, Czech
Republic, 5 Evolutionary Genomics Unit, Okinawa Institute of Science and Technology Graduate
University, Okinawa, Japan, 6 Department of Biochemistry, Faculty of Science, Charles University,
Prague, Czech Republic, 7 Department of Biochemistry and Microbiology, University of Chemistry
and Technology Prague, Prague, Czech Republic, 8 Department of Biological Sciences, Virginia Tech,
Blacksburg, Virginia, United States of America

* robert.hanus@uochb.cas.cz (RH); jitka.stafkova@uochb.cas.cz (JS)

doi.org/10.1371/journal.pbio.3003648

of Cambridge, UNITED KINGDOM OF GREAT
BRITAIN AND NORTHERN IRELAND

**Peer Review History:** PLOS recognizes the
benefits of transparency in the peer review
process; therefore, we enable the publication
of all of the content of peer review and
author responses alongside final, published
articles. The editorial history of this article is
available here: https://doi.org/10.1371/journal.
pbio.3003648

## Abstract

Termites produce the most diverse array of terpenoids among terrestrial metazoans,
comprising over 200 structures. However, their biosynthesis has not yet been eluci-
dated. Here, we identify a gene family which arose through a series of duplications
of geranylgeranyl pyrophosphate synthase in the common ancestor of Neoisoptera,
the terpene-producing termite lineage. We functionally characterized several proteins
from this rich GGPPS-like family as terpene synthases generating biologically rele-
vant sesqui- and diterpenes. These include the queen pheromone (3*R*,6*E*)-
nerolidol in *Embiratermes neotenicus* and the presumed precursor of polycyclic
defensive diterpenes (*E,E,E*)-neocembrene in *Nasutitermes takasagoensis*. We
report significant enrichment of transposable elements in the GGPPS-like genomic
loci, study the selection pressures acting in the evolution of the GGPPS-like family,
and highlight an amino acid site crucial for cyclization capacity and enantiospecificity
of the characterized enzymes. We conclude that we have identified an enzyme family
that facilitated the emergence of the remarkable richness of termite terpenoids.

## 1. Introduction

Termites hold the record among insects for the diversity of terpenoid secondary
metabolites they produce, with over 200 different monoterpenes, sesquiterpenes, and
diterpenes identified to date [1,2]. The occurrence of terpenoids is restricted to the
modern termite clade Neoisoptera and coincides with the emergence of the frontal

**Data availability statement:** *Embiratermes neotenicus* genomic data (draft genome assembly fasta, in silico annotated gene and protein models, annotation files) are deposited in the ASEP repository of the Czech Academy of Sciences (dataset ID 0640983, https://doi.org/10.57680/asep.0640983). Full gene sequences of *E. neotenicus* FPPS and GGPPS homologs are available in NCBI GenBank records PX512703-19. The transcriptomes of *Inquilinitermes inquilinus* and *Spinitermes trispinosus* are provided as NCBI TSA files under GenBank IDs GKIC00000000.1 and GLKH00000000.1. Caste-specific RNA-seq data used for differential gene expression analysis and GGPPS expression profiling in *E. neotenicus* are available as NCBI SRR archives (Biosamples SAMN53173298-SAMN53173329). The MS proteomics data have been deposited to the ProteomeXchange Consortium via the PRIDE partner repository (dataset PXD070647). Supplementary tables list, among others, sequences used in phylogenetic analyses and sequences of expressed proteins. Complete chromatographic and NMR data along with phylogenetic tree files and custom code related to transposable element analysis are available from OSF (doi: https://doi.org/10.17605/OSF.IO/RKDY9, https://osf.io/rkdy9).

**Funding:** This work was supported by the Czech Science Foundation (22-28470S, chemical analyses, bioinformatics, functional characterizations, proteomics; https://gacr.cz) to A.A., N.H., J.K., P.K., R.H., O.L., and J.Š., the Ministry of Education, Youth and Sports (LUC24118, mutagenesis, modeling; https://msmt.gov.cz) to A.A., N.H., J.Š., and R.H., the Charles University Grant Agency (371021; https://cuni.cz/uken-753.html) to N.H., and the US National Science Foundation (MCB1920925; https://www.nsf.gov) to D.T. We further acknowledge the Institute of Organic Chemistry and Biochemistry, CAS (RVO: 61388963; https://www.uochb.cz) for institutional support and the Ministry of Education, Youth and Sports of the Czech Republic for computational resources within e-INFRA CZ project (90254; https://msmt.gov.cz). The funders had no role in the study design, data collection and analysis, decision to publish or preparation of the manuscript.

gland, a unique exocrine organ and a synapomorphy of Neoisoptera. This gland is present in dispersing kings and queens, but most developed—and best studied—in soldiers, the caste specialized in colony defense [3,4].

The supposed ancestral function of termite terpenoids was to serve as defensive compounds, released in rich blends through the opening of the frontal gland on termite heads. Some of the defensive terpenoids were later co-opted for chemical communication, in line with the observations of pheromone evolution from defensive compounds in other insects [5]. Thus, terpenoids frequently act as alarm pheromones secreted from the frontal gland of irritated soldiers [6,7], but may also be synthesized in other exocrine organs and serve as sex pheromones and trail pheromones [8]. Finally, terpenoids may play the role of queen pheromones produced by reigning queens to signal their presence in the colony to the nestmates [9]. In contrast to the solid coverage of termite terpenoid diversity, their biosynthesis and its evolution remain to be elucidated. Because terpenoids have been recorded in the basal neoisopteran family Stylotermitidae [10] as well as in many other later diverging lineages, it is reasonable to hypothesize that their biosynthesis has a single origin in the common ancestor of Neoisoptera.

Terpenoid secondary metabolites usually originate from prenyl pyrophosphates, such as geranyl pyrophosphate ($C_{10}$, GPP), farnesyl pyrophosphate ($C_{15}$, FPP), or geranylgeranyl pyrophosphate ($C_{20}$, GGPP) giving rise to monoterpenes, sesquiterpenes, and diterpenes, respectively. The conversion of prenyl pyrophosphates into terpenes is catalyzed by terpene synthases (TPS), which cleave off the pyrophosphate and yield linear or cyclic terpene hydrocarbons, or the respective alcohols. Final terpenoid products may then arise through further modifications (oxygenations, cyclizations) by additional enzymes [11,12].

Like other animals, insects generally lack the "classical" TPS genes known from plants, bacteria, and fungi [13], except for a rare case of their gain via horizontal gene transfer [14]. Therefore, insects rely on alternative mechanisms to obtain terpenoids, such as sequestering them from the diet, acquiring them from microbial symbionts, or recruiting a completely unrelated enzyme family for TPS functions [15]. Nevertheless, the most common evolutionary route by which insects acquired the capacity for *de novo* terpenoid biosynthesis is through duplication and neofunctionalization of isoprenyl diphosphate synthase genes (IDS). This enzyme family is conserved in the primary metabolism across all domains of life and catalyzes the elongation of prenyl pyrophosphates via the sequential condensation of shorter pyrophosphates with isopentenyl pyrophosphate ($C_5$, IPP) [16,17]. Such origin of TPSs has so far been documented in a handful of species across Coleoptera, Lepidoptera, Hemiptera, and Diptera, where TPSs are paralogous either to geranyl and farnesyl pyrophosphate synthases (GPPSs and FPPSs [18–23]) or geranylgeranyl pyrophosphate synthases (GGPPS [24,25]). Taken together, these findings indicate that TPSs originated independently in multiple insect lineages through repeated IDS duplications and neofunctionalizations.

Here, we aim to elucidate the evolution of terpenoid biosynthesis in termites. Building on reports of independent origins of TPSs from IDSs in several insect clades, as

**Competing interests:** The authors have declared that no competing interests exist.

**Abbreviations:** BEB, Bayes Empirical Bayes; BUSCO, benchmarking universal single-copy orthologs; DMAPP, dimethylallyl pyrophosphate; EDTA, Extensive de novo TE Annotator; FDR, false discovery rate; FID, flame ionization detector; FPP, farnesyl pyrophosphate; FPPS, farnesyl pyrophosphate synthase; GC-MS, gas chromatography-mass spectrometry; GGPPS, geranylgeranyl pyrophosphate synthase; GPPS, geranyl pyrophosphate synthase; IBM, IPP-binding motifs; IDS, isoprenyl diphosphate synthase; IGMs, insect GGPPS motifs; IPP, isopentenyl pyrophosphate; LFQ, label-free quantitative; MS, mass spectrometry; NMR, nuclear magnetic resonance; ORFs, Open reading frames; PFC, preparative fraction collector; PTV, program temperature vaporization; SDC, sodium deoxycholate; SPME, solid-phase microextraction; TEs, transposable elements; TOF, Time-of-Flight; TPM, transcripts per kilobase million; TPS, terpene synthase.

well as on the exclusive occurrence of terpenoids in Neoisoptera, we hypothesize that terpenoid biosynthesis evolved in the common ancestor of Neoisoptera through duplication/multiplication and neofunctionalization of IDSs. To capture the structural and functional diversity of termite terpenoids, we selected two model species from different subfamilies within Neoisoptera. Our primary model is the South American termite *Embiratermes neotenicus* (Termitidae: Syntermitinae), in which we recently identified the sesquiterpene alcohol (3*R*,6*E*)-nerolidol as the queen pheromone [9]. The second model is the Japanese species *Nasutitermes takasagoensis* (Termitidae: Nasutitermitinae) belonging to a genus whose soldiers produce a rich blend of monoterpenes and polycyclic oxygenated diterpenes presumably derived from the monocyclic precursor (3*E*,7*E*,11*E*)-neocembrene [1,2,26]. Previously, eight GGPPS homologs have been reported as present and upregulated in the frontal gland of soldiers in this species [27]. Although GGPPS gene multiplicity has been tentatively linked to terpenoid production in termites [27,28], the hypothesis has not been experimentally verified.

We report that GGPPS exists as a single-copy gene in basal termites, but its duplication or multiplication is a ubiquitous feature of Neoisoptera, with paralog numbers reaching up to 15 in *E. neotenicus*. This pervasive multiplicity strongly indicates that the initial duplication event occurred concurrently with the frontal gland emergence in their common ancestor. We functionally characterize selected GGPPS homologs from the two species and demonstrate that some of them act as TPS enzymes, converting prenyl pyrophosphate substrates into terpenoids known from the respective species, or biosynthetic precursors thereof. We conclude that we have identified a family of TPS enzymes that facilitated the emergence of the remarkable richness of terpenoid secondary metabolites in termites. Furthermore, we bring evidence for possible involvement of transposable elements (TEs) in shaping the GGPPS-like gene family and outline how selection pressures may have affected its evolutionary diversification. Finally, we examine the structure-function relationship of termite TPS enzymes and identify an amino acid crucial for their capacity of terpene cyclization and enantioselective biosynthesis.

## 2. Results

### 2.1. Terpenoid chemistry in *E. neotenicus* queens and *N. takasagoensis* soldiers

To obtain a comprehensive image of the terpenoids produced by the two model termites, we first analyzed body washes from selected castes of both species using gas chromatography coupled with mass spectrometry (GC-MS). In *E. neotenicus*, substantial amounts of terpenes were only present in neotenic queens; the dominant compound was (6*E*)-nerolidol, followed by (3*E*,7*E*,11*E*)-neocembrene, along with minor or trace quantities of α- and β-farnesene, β-springene and two additional unidentified diterpenes (Figs 1A and S1, S2).

In soldiers of *N. takasagoensis*, we detected a rich and abundant blend of terpenes comprising two monoterpenes—(+)-α-pinene and (+)-limonene—and eight diterpenoids (Figs 1B and S3–S5). Three of the most prominent diterpenoids, each bearing

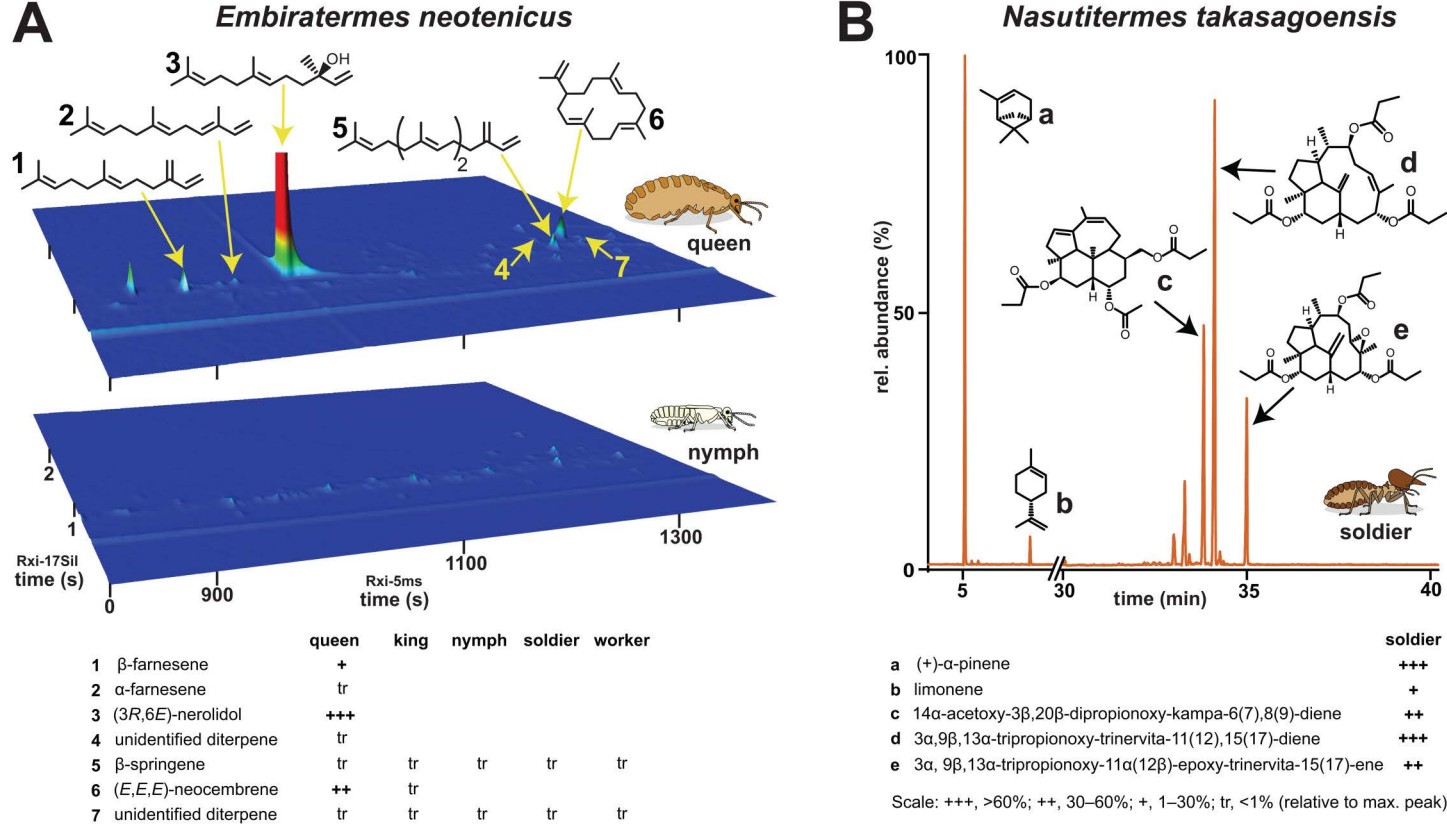

**Fig 1. Terpenoid chemistry of the two model termite species. A.** Comprehensive gas chromatography (GC×GC) analysis showing terpenes identified in the body washes of neotenic queens of *E. neotenicus* and compared to 4th stage female nymphs. Chromatograms of other castes are provided in S1 Fig, mass spectra of the identified terpenes in S2 Fig. **B.** GC chromatogram of terpenes identified in body washes of *N. takasagoensis* soldiers. All chromatograms (A and B) show the responses to diagnostic ions for mono-, sesqui-, and diterpenes ($m/z = 68 + 69 + 93 + 204 + 272$). Comparison with a chromatogram of worker body wash is shown in S3 Fig, mass and NMR spectra of the detected terpenes are shown in S4–S8 Figs. Complete chromatographic data are available from OSF (https://osf.io/rkdy9).

three or four oxygenated functional groups, were fully characterized using nuclear magnetic resonance (NMR) as 14α-acetoxy-3β,20β-dipropionoxy-kampa-6(7),8(9)-diene, 3α,9β,13α-tripropionoxy-trinervita-11(12),15(17)-diene, and 3α,9β,13α-tripropionoxy-11α(12β)-epoxy-trinervita-15(17)-ene (S6–S8 Figs), following termite diterpene nomenclature [29].

## 2.2. An FPPS gene duplication predates the occurrence of termites

We did not retrieve any sequences similar to plant, fungal, or bacterial TPSs in the transcriptome assemblies of *E. neotenicus* and *N. takasagoensis*, nor in the *E. neotenicus* genome, which we assembled for this study (genome assembly statistics are provided in S1 Table). No duplication was found in genes coding for subunits of decaprenyl-diphosphate synthase, a long-chain IDS. We then queried the datasets for short-chain IDS homologs. Our search confirmed the presence of three previously reported FPPS sequences in *N. takasagoensis* transcriptome [27] and revealed an FPPS tandem duplication in *E. neotenicus* genome. A phylogenetic analysis of FPPS homologs from selected cockroach and termite species suggested that an FPPS gene duplication is present in all Blattodea (S9A Fig). An inspection of amino acid sequence alignments showed that the more conserved blattodean FPPS clade, containing one and two sequences of

PLOS Biology

*E. neotenicus* and *N. takasagoensis*, respectively, presents all main features typical of the ancestral FPPS enzymes, such as the so-called first and second aspartate-rich motifs (FARM and SARM, sequence DDxxD) implicated in binding of magnesium ions in the active site of the enzyme [30,31]. We therefore designated this clade as canonical FPPSs. By contrast, in the divergent clade which we refer to as FPPS-like, FARM and SARM are disrupted to an extent which does not allow for M(II) ion coordination (S9 and S10 Figs). The distinction between FPPS- and FPPS-like proteins was confirmed by *in vitro* assays with recombinant EneoB from *E. neotenicus*, a member of the FPPS clade, which presented activities typical of insect FPPS proteins. No IDS or TPS activity was detected with the FPPS-like protein from the same species, EneoA (S9D Fig). The combination of phylogenetic analysis, protein active site modeling, and biochemical assays let us conclude that the FPPS-like homologs are unlikely to be involved in terpenoid biosynthesis in termites.

### 2.3. Neoisoptera possess an expanded family of GGPPS-like genes

Next, we searched the *E. neotenicus* transcriptome and genome for GGPPS homologs and identified multiple full-length GGPPS sequences. Interestingly, we also discovered numerous partial GGPPS homologs lacking one or more exons and located in genomic regions extensively soft-masked by RepeatMasker. An inspection of the unmasked genome revealed as many as 28 highly homologous sequences with conserved gene organization (Figs 2A and S11), most of which were expressed in one or more *E. neotenicus* castes. Based on gene integrity (six protein-coding exons) and the presence of indels or nonsense mutations, we classified 15 of these GGPPS homologs as genes and 13 as pseudogenes.

We then reconstructed the GGPPS nucleotide phylogeny in the context of publicly available sequences from other insects, including two basal termites and three Neoisoptera (*Stylotermes halumicus*, *E. neotenicus*, and *N. takasagoensis*). GGPPS occurs as a single-copy gene in basal termites and other insects, whereas multiple paralogs were found in Neoisoptera, including *S. halumicus* from the very basal neoisopteran family Stylotermitidae (Fig 2B). Neoisoptera possess one paralog clustering with the single-copy genes of basal termites (denoted as clade I), presumably representing the canonical GGPPS enzyme. The remaining paralogs form an expanded monophylum, further referred to as the GGPPS-like gene family, comprising three sequences from *S. halumicus*, six from *N. takasagoensis,* and 14 from *E. neotenicus*. The GGPPS-like clade resolves into three branches, denoted II–IV (Fig 2A, 2B).

We further verified the hypothesis that the GGPPS multiplication is specific to Neoisoptera by phylogenetic analysis of GGPPS amino acid sequences including 7 basal termite species and 19 Neoisoptera (S12A Fig, S6 Table). The presumed canonical GGPPS branches (paraphyletic clade I) comprised sequences from cockroaches, basal termites, and Neoisoptera, while the divergent GGPPS-like branch consisted exclusively of neoisopteran sequences and formed a monophylum sister to neoisopteran GGPPSs. All Neoisoptera species were represented in the GGPPS-like branch, subdivided into well-supported clades II–IV. Thus, the initial GGPPS duplication occurred in the ancestor of Neoisoptera and was followed by subsequent duplication events. Additionally, we broadened the sampling for the phylogenetic analysis and included sequences from *Heliconius* butterflies, in which two GGPPS-like TPSs were reported, as well as several insect GGPPSs with confirmed IDS function. The reconstructed tree clearly shows that GGPPS duplications arose independently in Neoisoptera and in *Heliconius* butterflies (S12B Fig, S6 Table).

Finally, we surveyed all publicly available sequences homologous to GGPPS across termites, including partial TSA transcripts, and mapped them on a phylogenetic tree of termites together with the records of terpenoid secondary metabolites in Neoisoptera (Fig 3, S7 Table). We obtained a clear picture of GGPPS evolution: while only GGPPS could be retrieved from the basal lineages represented by 8 species from three families, at least some partial GGPPS-like sequences were identified in all 46 included Neoisoptera from seven families and 12 subfamilies of Termitidae.

### 2.4. GGPPS and GGPPS-like genomic loci are rich in TEs

In *E. neotenicus*, the observed extensive duplication in GGPPS and GGPPS-like genes occurs alongside an increased frequency of low-complexity regions in the given loci. To elaborate on this observation, we performed a set of enrichment

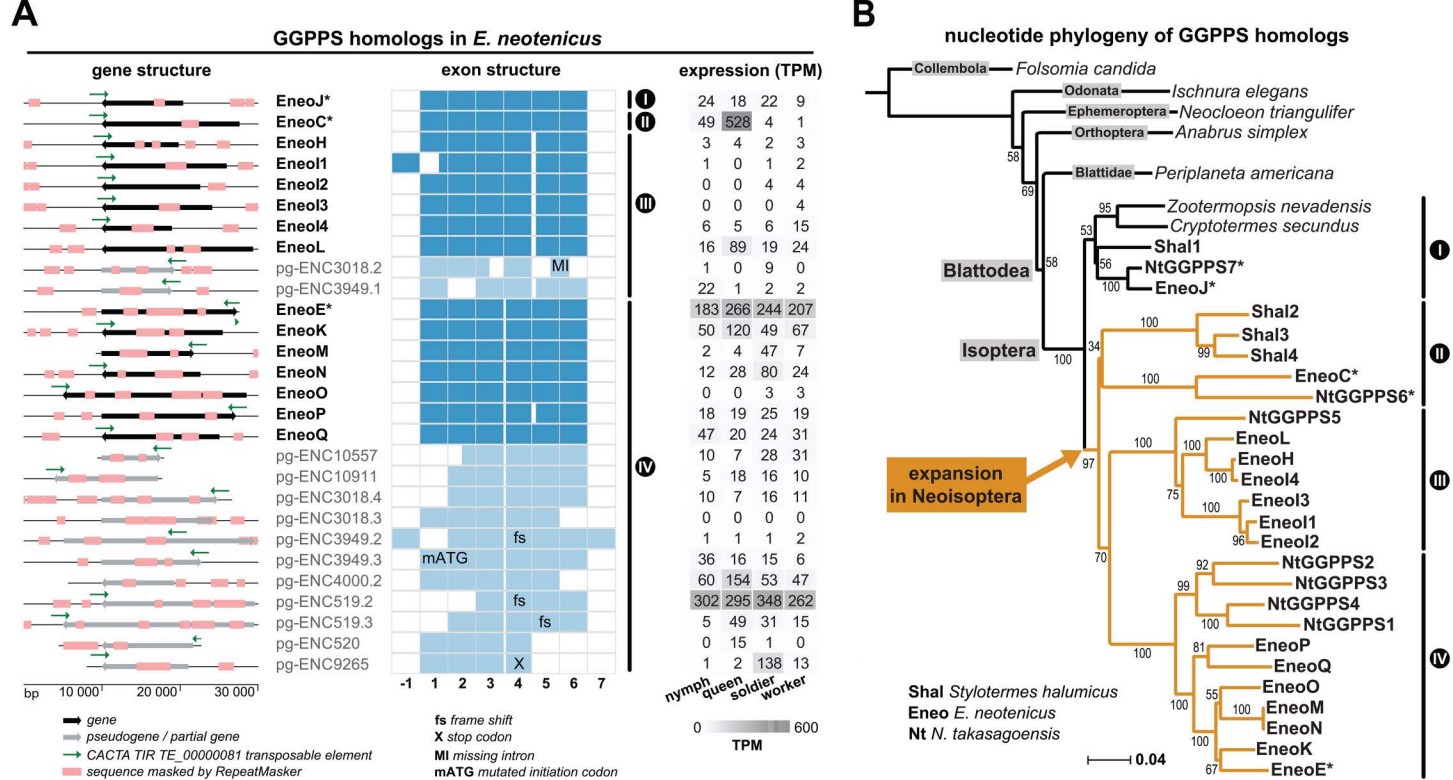

**Fig 2. Diversity, gene structure, expression, and phylogeny of GGPPS homologs in termites. A.** GGPPs homologs identified in the genome of *E. neotenicus,* their gene (left) and exon (middle) structure, and expression in different castes (right). Gene structure: genes are marked with black arrows, pseudogenes with gray arrows, regions masked by RepeatMasker at genome-wide scale are highlighted in pink. Sequences corresponding to CACTA TIR TE_00000081 element are represented by green arrows also showing the TE orientation. Exon structure: alternative 5′ exons in EneoH and pgENC3949.2 located upstream of the canonical exon 1 are marked as "−1". Expression profiles: TPM normalized read counts (sum of counts for head and thorax + abdomen samples) are shown as a heatmap based on mRNA-seq of *E. neotenicus* 4th stage female nymphs, neotenic queens, workers, and soldiers. **B.** Phylogenetic tree reconstructed from protein-coding nucleotide sequences showing the GGPPS-like gene family expansion in Neoisoptera. The tree was generated using the Neighbor-Joining method with 10,000 bootstrap replicates. Neoisopteran gene abbreviations are set in bold. Asterisks mark functionally characterized enzymes. Accession numbers of the studied sequences are provided in S4 and S5 Tables. The tree file is available from OSF (https://osf.io/rkdy9).

analyses using genome-scale TE prediction with the Extensive de novo TE Annotator (EDTA) pipeline. The mean TE density observed in a 100 kb window surrounding the GGPPS-like genes and pseudogenes was 0.65 which corresponds to 150% of the average density 0.43 observed in the genomic windows of the same size generated within 1,000 random permutations or 0.42 in regions surrounding protein-coding genes. An even lower TE density of 0.26 was observed in the FPPS-like locus (S13A Fig). The high observed TE density in the GGPPS-like loci corresponds mostly to TEs located next to the genes while exhibiting a periodic distribution near genes and proximal paralogs (S13B Fig). As documented by the TE family representation analysis, no single TE type was found to be specifically enriched in the GGPPS-like loci (S13C Fig).

As all the paralogs retained an unspliced gene structure with fully conserved exon-intron boundaries, we focused on class II DNA TEs—elements that transpose directly as DNA without splicing [33]. Upon running RepeatMod-eler at the whole-genome level, we observed multiple DNA TE-like motifs in loci surrounding the GGPPS-like genes and pseudogenes, consistent with the increased occurrence of masked regions. To improve prediction accuracy, we restricted our analysis to 27 contigs harboring sequences which showed >80% homology to previously identified

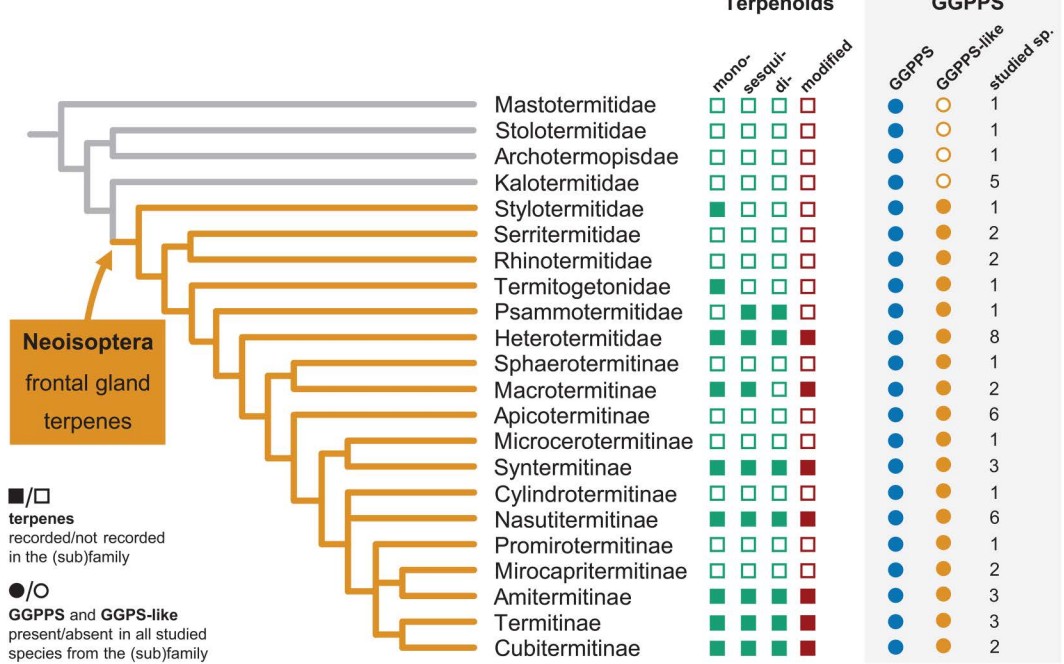

**Fig 3. Evolution of terpenoid biosynthesis in termites.** Production of mono-, sesqui-, and diterpenes, and their secondary derivatives (containing complex polycycles and oxygenated moieties), presence of GGPPS and GGPPS-like sequences, mapped on termite phylogeny of nontermitid families and termitid subfamilies. Phylogeny adopted and simplified from Hellemans and colleagues [32], Hodotermopsidae and Serritermitidae are excluded from the scheme due to the absence of relevant sequence data. Positive records of terpenes denote previous identifications of any terpenoid secondary metabolite in at least one species from the given (sub)family, based on [1,2], and the recent report of terpenes in Stylotermitidae as the most basal Neoisoptera [10]. The presence or absence of GGPPS and GGPPS-like sequences in individual termite species was determined based on GGPPS phylogeny and/or the presence of motifs previously identified as characteristic for insect GGPPSs. Numbers show the sampling size in each (sub)family. The termite species and relevant datasets are listed in S7 Table.

GGPPS-like orthologs and at least two consecutive exons. Among several TEs revealed by the TE annotation pipeline, TE_00000081—a class II transposon from the CACTA TIR family—was found in all but two loci. This TE was oriented in the opposite direction relative to the gene and was located in the region surrounding exon 6 (Figs 2A and S13).

### 2.5. Identification of proteins with GGPPS activities in *E. neotenicus* and *N. takasagoensis*

First, we functionally characterized the proteins from clade I expected to act as GGPPS enzymes (EneoJ in *E. neotenicus* and NtGGPPS7 in *N. takasagoensis*). Indeed, both proteins showed IDS activity with IPP in combination with all allylic substrates tested: they produced GGPP from IPP and FPP; IPP and GPP generated both FPP and GGPP; upon incubation with IPP and DMAPP, only GPP was detected (S14 Fig). No TPS activity was observed in EneoJ or NtGGPPS7 for any of the substrates tested (GPP, FPP, and GGPP).

### 2.6. Two *E. neotenicus* GGPPS-like proteins show TPS activity and generate queen terpenoids

Among the 14 *E. neotenicus* GGPPS-like paralogs, we selected EneoC and EneoE, highly expressed in neotenic queens according to RNA-Seq analysis shown in Fig 2C. Upon incubation of EneoC with FPP, a single product was generated, identified by GC-MS and chiral GC as pure (3R,6E)-nerolidol (Fig 4A). Additionally, EneoC presented IDS activity by

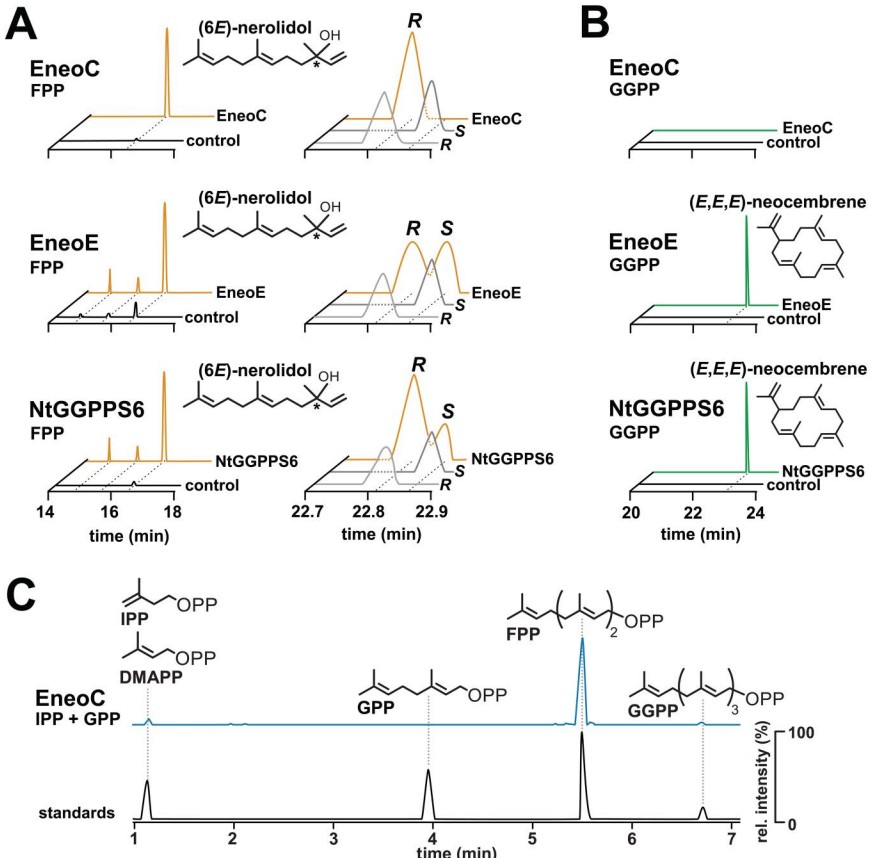

**Fig 4. Functional characterization of GGPPS-like enzymes from *E. neotenicus* and *N. takasagoensis*. A.** GC chromatograms of SPME samples obtained from the headspace upon EneoC, EneoE, and NtGGPPS6 incubation with FPP. Left chromatograms show separation on a semi-nonpolar column, right chromatograms depict chiral separation of (6*E*)-nerolidol on a chiral column. Enzymatic assays shown in yellow, controls without enzymes in black, standards of (6*E*)-nerolidol enantiomers in gray. **B.** GC chromatograms of SPME samples obtained from the headspace upon EneoC, EneoE, and NtGGPPS6 incubation with GGPP. Enzymatic assays shown in green, controls without enzymes in black. **C.** HPLC chromatogram demonstrating the IDS activity of EneoC. The purified enzyme was incubated with IPP in combination with DMAPP, GPP, or FPP. Only the substrate combination giving rise to prenyl pyrophosphate product (FPP) is shown (blue), along with standards (black). The chromatogram visualizes the selected m/z 245.00, 313.06, 381.12, and 449.19. Complete chromatographic data are available from OSF (https://osf.io/rkdy9).

generating FPP upon incubation with IPP and GPP (Fig 4C). No other substrates were accepted by EneoC (GPP or GGPP in TPS assays, IPP combined with DMAPP, FPP, or GGPP in IDS assays).

EneoE accepted FPP in TPS activity assays, generating racemic (6*E*)-nerolidol, and converted GGPP into (3*E*,7*E*,11*E*)-neocembrene (Fig 4A, 4B). The enzyme showed no IDS activity. Both (3*R*,6*E*)-nerolidol and (3*E*,7*E*,11*E*)-neocembrene represent queen-specific terpenoids in *E. neotenicus* (Fig 1A), the former being known as *E. neotenicus* queen pheromone [9]. Thus, we consider EneoC an enantiospecific nerolidol synthase and EneoE a neo-cembrene synthase.

Having identified EneoC as capable of producing the *E. neotenicus* queen pheromone, we wanted to confirm its upregulation in queens compared to nymphs. Indeed, in differential gene expression analysis of RNA sequencing data from nymphs and queens, EneoC was by far the most upregulated of all GGPPS-like genes in queen samples from thorax and abdomen ($\log_2$ fold change queen versus nymph = 9.9, padj = $1.8 \times 10^{-16}$) while also showing the highest transcript abundances (S15 Fig, S8 Table). Importantly, EneoC transcript upregulation was clearly reflected by proteomic data. Among

all detected IDS homologs, EneoC was present as the most abundant protein and showed the strongest enrichment in queen abdomen and thorax compared to nymphs ($\log_2$ fold change queen versus nymph = 10.6; S15 Fig, S9 Table). In summary, both transcriptomic and proteomic analyses supported the biological relevance of EneoC for queen pheromone biosynthesis.

### 2.7. *N. takasagoensis* GGPPS-like protein NtGGPPS6 is a neocembrene synthase

All six GGPPS-like genes were reported as upregulated in *N. takasagoensis* soldier heads, the biosynthetic site of terpenoid defensive compounds [27]. We expressed these in the *Saccharomyces cerevisiae* strain JWY501 which is specifically engineered to accumulate GGPP. In the NtGGPPS6-expressing strain, we detected high amounts of (3*E*,7*E*,11*E*)-neocembrene, while no significant terpenoid production was observed in any other strain (S16 Fig). Subsequently, we expressed the six GGPPS-like genes in *Escherichia coli* and used the purified proteins in IDS and TPS assays. Only recombinant NtGGPPS6 showed any activity *in vitro*, producing a mixture of (3*R*,6*E*)- and (3*S*,6*E*)-nerolidol enantiomers from FPP (Fig 4A), and (3*E*,7*E*,11*E*)-neocembrene from GGPP (Fig 4B). No IDS activity was observed. As only monoterpenes and diterpenes are present in the soldier extract, we assume that *in vivo*, NtGGPPS6 does not generate nerolidol, and acts instead as a neocembrene synthase producing the precursor of polycyclic diterpenes.

### 2.8. Sequence motifs within termite GGPPS gene family and selection analysis

We examined whether the divergence of GGPPS-like genes involves substitutions within motifs typical of GGPPS proteins. We defined such insect GGPPS motifs (IGMs) based on patterns generated for canonical insect GGPPS sequences by MEME, and we combined them with the IPP-binding motif prediction used in our earlier study [34]. We observed that only FARM and IGM 3 motifs are conserved in termite GGPPS-like proteins (S17 Fig).

To understand how selection pressures may have influenced the evolution of TPS function in Neoisoptera, we performed selection analysis on a phylogeny of selected full-length termite GGPPS and GGPPS-like coding sequences (Fig 5A, S10 Table) with the HyPhy RELAX. Our analysis showed that selection pressures have overall intensified along branches in the GGPPS-like clade containing the neoisopteran TPSs (k = 11.1, $p < 10^{-4}$). The partitioned descriptive model fitted to each branch class indicated a general increase in the intensity of diversifying selection along GGPPS-like clade branches compared to the canonical GGPPS reference branches (S11 Table, ω = 22.72 versus 3.58). Additionally, a lower degree of constraint from purifying selection acting on the GGPPS-like versus canonical GGPPS clades can be inferred from the substantial decrease in the proportion of codons under negative selection (S11 Table, 33.34% in GGPPS-like sequences versus 97.06% in canonical GGPPSs) and a concurrent emergence of a category of neutrally evolving codons in the test branch class (S11 Table, 64.45% versus 0).

We next tested all branches within the termite GGPPS-like clade for evidence of positive selection affecting the diversification of the clade by applying the aBSREL (adaptive branch-site random effects likelihood) procedure in HyPhy to our dataset. Three internal branches were significantly likely to have experienced positive diversifying selection (Holm-Bonferroni-corrected *p*-values < 0.05). Two of these occur within the GGPPS-like clade II containing EneoC and NtGGPPS6. We designated the internal nodes delimiting these branches 1, 2, and 3 (Fig 5A), and used PAML to reconstruct their ancestral sequences (S12 Table). PAML was also employed to detect specific amino acid sites affected by positive selection (ω > 1) using the Bayes Empirical Bayes (BEB) method. For the branch connecting nodes 2 and 3, posterior probability higher than 0.9 was recorded in four sites corresponding to the following putative amino acid changes: P23S, Q26G, S59G, and N90S (Fig 5B–5D). Two additional sites (Q21K, S60I) showed BEB values higher than 0.8. These sites were retained almost completely in both NtGGPPS6 (K21, S23, G26, G59, G60, and S90) and EneoC (K21, S23, G26, C59, I60, and S90). No site with a BEB value exceeding 0.9 was detected on the branch leading to node 2. A structure model of ancestral sequence 2 was then generated using AlphaFold 3 and compared to NtGGPPS6 (Fig 5B, 5C), showing that all listed sites are located

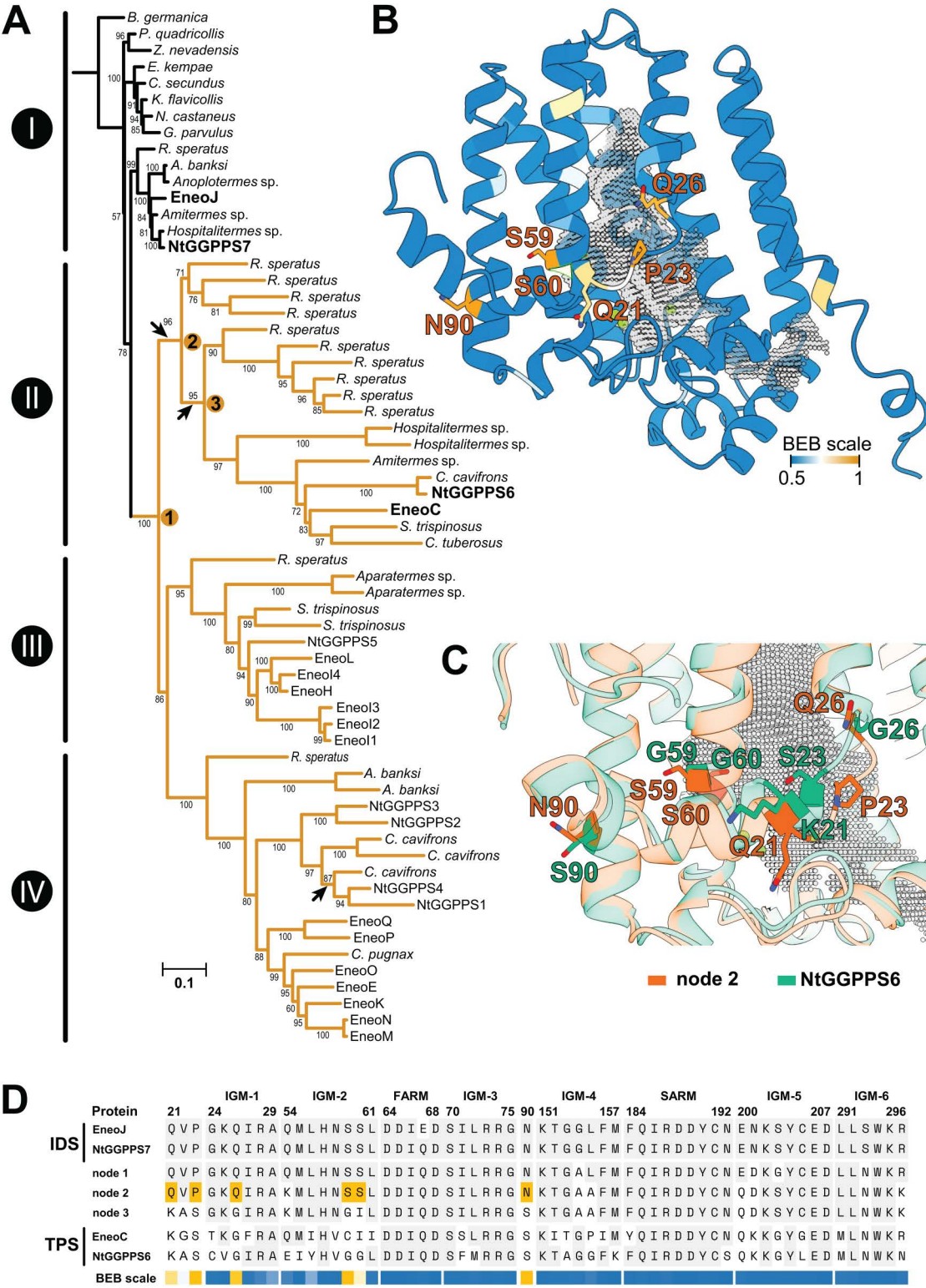

**Fig 5. Selection analysis of termite GGPPS homologs. A.** Maximum-likelihood phylogenetic tree based on codon alignment of sequences listed in S10 Table, reconstructed using IQ-Tree and substitution model JTT+F+I+G4, ultrafast bootstrap values (1,000 replicates) shown as node support. RELAX analysis was performed with GGPPS sequences (clade I, reference branches) and GGPPS-like clades II–IV (yellow highlighting, test branches)

showing increased intensity of positive selection in GGPPS-like clades. The arrows point to internal branches along which positive selection has likely occurred according to aBSREL analysis. Ancestral sequences were reconstructed for nodes 1–3 (S12 Table). Functionally characterized extant sequences relevant for the selection analysis are set in bold. **B.** Structure model of ancestral sequence at node 2 with amino acid residues colored with respect to Bayes Empirical Bayes posterior probabilities of the site having experienced positive selection (0.5–1, sites with BEB values > 0.8 are labeled). Active site cavity is filled with gray mesh, $Mg^{2+}$ ions shown as green spheres. **C.** Overlay of modeled structures of NtGGPPS6 (green) and ancestral sequence 2 (orange) focusing on the amino acid sites highlighted in B. **D.** Abridged alignment of amino acid sequences of extant GGPPS and TPS sequences in target species and ancestral sequences 1–3 showing residues at insect GGPPS motifs 1–6 (IGMs), FARM, and SARM. Amino acids identical to GGPPS sequences are shaded gray. BEB values at individual sites for the branch 2–3 are shown as in B, sites in node 2 sequence with BEB values > 0.8 are shaded yellow. The tree file is available from OSF (https://osf.io/rkdy9).

near the active site cavity opening. Three of these (Q21K, P23S, and Q26G) lie on a loop adjacent to the opening, a region with greater conformational flexibility compared to the prevailing α-helical secondary structure.

## 2.9. Structural features underlying cyclization capacity and enantioselectivity of termite TPS

We modeled structures of proteins with known activity (EneoC, EneoE, EneoJ, NtGGPPS6, and NtGGPPS7), and mapped the IGM motifs onto the structures. Overall, all protein structures aligned very well as exemplified by EneoJ and EneoC models (Fig 6C). Interestingly, a distinct pre-cyclization GGPP substrate conformation was modeled with EneoE protein, contrasting with a rather linear GGPP positioning in the active site of NtGGPPS6 (Fig 6D). To evaluate the proximity and potential significance of individual amino acid changes for protein enzymatic activities, we focused on the less divergent clade II GGPPS-like TPS enzymes (EneoC, NtGGPPS6). Specifically, we searched for variable amino acids located near the FARM and SARM motifs within the highly polar part of the active site where the isoprenoid carbocation is formed and transferred to an IPP molecule in IDS enzymes, taking IGM motif substitutions into account. We noticed an interesting site within IGM 4, a motif located in helix 6, ~8 Å from FARM (Fig 6C–6D). Thr152 is conserved in NtGGPPS6 and all IDS enzymes as a part of the KT motif (Fig 6A), and replaced by isoleucine in EneoC. We hypothesized that Thr152 might be important for carbocation stabilization which would aid GGPP cyclization into neocembrene by NtGGPPS6. Indeed, the substitution of this amino acid in the T152I mutant of NtGGPPS6 abolished neocembrene production from GGPP and instead lead to the formation of the linear diterpene alcohol geranyllinalool, a $C_{20}$ homolog of nerolidol. In addition, upon incubation of T152I NtGGPPS6 with FPP, we observed a significant shift in the enantioselectivity of (6*E*)-nerolidol biosynthesis. While the wild type produced 65% of the *R* enantiomer, this proportion increased to 91% in the mutant enzyme, thus nearing the full enantioselectivity of EneoC (Fig 6B, S13 Table).

## 3. Discussion

A novel biosynthetic capacity emerged in termites of the modern lineage Neoisoptera, enabling them to produce a remarkably broad scope of terpenoid secondary metabolites. The goal of this study was to unveil the evolutionary background of this innovation by identifying TPS enzymes, generating either the final terpenoids or precursors thereof. We provide evidence for TPS activity within a family of GGPPS-like enzymes, which evolved by a duplication of a GGPPS gene in the ancestor of Neoisoptera and a subsequent expansion accompanied by the gain of a novel function. The identification of termite TPS enzymes as members of a GGPPS-like gene family adds another instance to the list of independent acquisitions of TPS function via the duplication and neofunctionalization of ancestral IDS (FPPS or GGPPS) genes previously reported in several lineages of insects and in fungi [18–25,35]. In fact, the classical TPS gene family in plants and microorganisms has itself been hypothesized to have arisen from an IDS ancestor [13,36], possibly making its origin the most ancient of the recurring IDS neofunctionalization events.

Our data show that GGPPS-like family evolved in the common ancestor of Neoisoptera, suggesting its concurrent origin with the terpene-producing frontal gland, a key neoisopteran synapomorphy. A broader examination of published data shows that the evolutionary origin of terpene synthases in the ancestor of Neoisoptera aligns with the presence of terpenoid secondary metabolites across termite phylogeny, as both features are first recorded in the basal neoisopteran

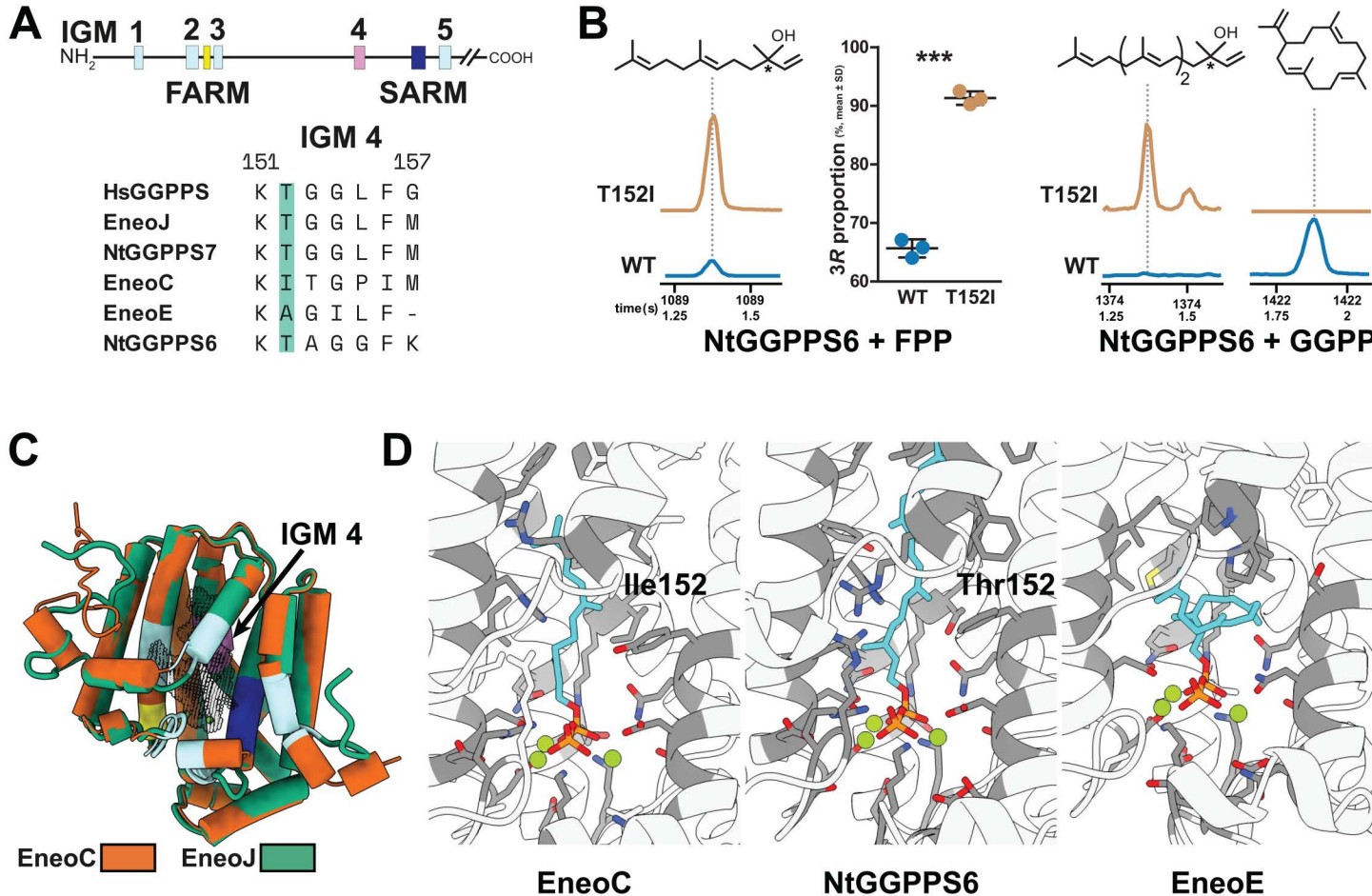

**Fig 6. Comparison of overall structures and active sites of termite GGPPS and GGPPS-like proteins. A.** Location of insect GGPPS motifs (IGMs) and aspartate-rich motifs (FARM and SARM) in GGPPS and GGPPS-like sequences; IGM 4 motif alignment in functionally characterized termite proteins and the human HsGGPPS, position 152 highlighted. **B.** TPS activities of T152I NtGGPPS6 mutant and the native protein (WT) documented by GC chromatograms of SPME samples. Left, (6*E*)-nerolidol generation from FPP and the plot of (3*R*,6*E*)-nerolidol enantiomer proportion determined by chiral analysis (mean values ± SD, source data in S13 Table). Right, (6*E*,10*E*)-geranyllinalool and (*E,E,E*)-neocembrene generation from GGPP. **C.** Aligned structural models of EneoJ (GGPPS, green) and EneoC (nerolidol synthase, orange). Colored regions correspond to IGM 1–3 and IGM 5 (light blue), IGM 4 (pink), FARM (yellow), and SARM (dark blue). Mg²⁺ ions are shown as green spheres. The active site cavity is represented by a gray mesh. **D.** Active site cavity with fitted substrates in cyan (FPP in EneoC, GGPP in NtGGPPS6 and EneoE) and amino acids close to the substrate in dark gray. Mg²⁺ ions are shown as green spheres. Complete chromatographic data are available from OSF (https://osf.io/rkdy9).

genus *Stylotermes* (Fig 3). The GGPPS-like genes remained present in the genomes of all later branching clades over roughly 100 million years of their diversification (according to dating by Buček and colleagues [37]), even though in some extant lineages the terpenoids themselves seem to be missing. In later branching lineages of Neoisoptera, a trend toward higher terpenoid complexity can be observed, presumably reflecting the involvement of additional enzymes acting on TPS-produced skeletons (Fig 3). For example, the oxygenated polycyclic diterpenes typically found in the modern lineage Nasutitermitinae, such as the three highly complex structures identified here in *N. takasagoensis* (Fig 1B), represent nearly one half of the termite terpenoid repertoire [1,2]. They are expected to arise via subsequent modifications of the TPS-produced (3*E*,7*E*,11*E*)-neocembrene [1,26] and we propose that the most intuitive candidates for further cyclizations and oxygenations are cytochromes P450, some of which are reported as conspicuously upregulated in the frontal glands of *Nasutitermes* soldiers [12,27].

In eusocial insects, a gene family expansion accompanied by a caste-biased expression pattern of newly emerging paralogs, as observed here in the case of GGPPSs, represents a feature often associated with caste diversification [38–41]. Multiple examples have been accumulated showing that duplications and neofunctionalizations of gene sets and their differential use in individual castes drive social insect phenotypic plasticity, e.g., in genes involved in endocrine signaling, chemosensation, metabolism, or digestion [28,42–45].

The emergence of a gene family and its further expansion can be mediated by different mechanisms, which are not necessarily shared in IDS-like gene families in individual insect lineages. For example, the genomic organization of pan-Blattodean FPPS-like genes in termites is suggestive of origin by tandem duplication, which also appears to have been a major driver of FPPS-like gene family expansion in stink bugs [23]. Alternatively, gene duplication can be mediated by TEs [46], examples including non-TPS genes participating in terpenoid biosynthesis in roses [47]. Our data also suggest the potential involvement of TEs in shaping the expansion of GGPPS-like family in *E. neotenicus*. The observed mean TE density in the near proximity of the GGPPS-like genes and pseudogenes is significantly higher compared to random genomic regions as well as to loci surrounding protein-coding genes. A specific TE, TE_00000081, was consistently associated with focal loci in *E. neotenicus*, and sequences homologous to this element are also present in GGPPS-like transcripts from *N. takasagoensis* and in *Reticulitermes speratus* GGPPS-like genomic locus (S13 Fig). We thus propose that GGPPS gene duplications in Neoisoptera were frequently concurrent with transposition events.

We have identified three TPS enzymes which all produce (6*E*)-nerolidol from FPP *in vitro*. Out of these, only EneoC is an enantiospecific TPS generating pure (3*R*,6*E*)-nerolidol, which acts as a queen pheromone in *E. neotenicus*. *EneoC* gene has been found as highly upregulated in queens at transcript and protein levels, both in an inter-caste comparison and relative to other GGPPS-like genes, supporting its biological role in queen pheromone production. Interestingly, to our knowledge, EneoC is the only known terpene synthase producing exclusively the *R* enantiomer of nerolidol. The only other enzyme reported to generate enantiomerically pure (3*R*,6*E*)-nerolidol is a maize enzyme which, however, also produces additional sesquiterpenes [48]. The other two TPSs which we functionally characterized, NtGGPPS6 and EneoE, presumably act as (3*E*,7*E*,11*E*)-neocembrene synthases *in vivo*, thus generating the putative precursor of defensive compounds in *N. takasagoensis* and a queen-specific volatile in *E. neotenicus*. In addition to TPSs, we have also identified canonical GGPPSs in both *E. neotenicus* and *N. takasagoensis* corresponding to typical insect class III GGPPS enzymes [49–51].

To obtain a comprehensive view of terpenoid biosynthesis in termites, it would be required to identify the enzymes responsible for production of monoterpenes. Therefore, one of the selected models was *N. takasagoensis*, whose defensive compounds are rich in pinene and limonene. To our dismay, none of the recombinant *N. takasagoensis* enzymes showed monoterpene synthase activity. Recognizing the abundance of GGPPS-like paralogs in Neoisoptera and the complexity of their correct annotation, we propose that a successful TPS activity screen requires a high-quality candidate list based on manually annotated genomic and transcriptomic sequence data. Alternatively, we cannot rule out that the biosynthesis of monoterpenes is independent of the GGPPS-like expansion, though this option is rather unlikely given the concurrent emergence of monoterpenes and GGPPS-like genes in the most basal Neoisopteran lineage Stylotermitidae [10].

Genetic novelty provides the substrate for evolutionary processes, and our analyses suggest that selection has shaped the diversification of the Neoisopteran GGPPS-like gene family. We show that the intensity of diversifying selection was higher along branches in the GGPPS-like clades compared to the GGPPS clade and that the GGPPS-like clades are less constrained by purifying selection, both of which facilitates specialized adaptation in general [38]. Furthermore, the basal branches within GGPPS-like clade II, which contains TPS enzymes EneoC and NtGGPPS6, likely experienced episodic positive diversifying selection. Positive selection-driven diversification of terpenoid biosynthesis has been reported in plants [52–54] and we are presenting here an analogous phenomenon in termites. In an early branch of the same clade II, we have identified six amino acids sites with evidence of positive selection corresponding to the following substitutions along the branch: P23S, Q26G, S59G, N90S, Q21K, and S60I (Fig 5B–5D). Ser59 and Ser60 are associated with

substrate and product chain-length specificity in type III GGPPS [55,56]. In fungal GGPPS-like TPSs, the S60C mutation shifts product specificity from the alcohol (*E*)-nerolidol to (*E,E*)-α-farnesene [35]. Gln21 typical for insect GGPPS enzymes is changed in termite TPSs (Q21K) and retained in both TPSs of the butterfly *Heliconius melpomene* [24]. At a corresponding position, fungal GGPPS sequences of *Melamspora* share Asn, while an N21I change is present in GGPPS-like TPSs from the same species [35]. Thus, in two out of three lineages with confirmed GGPPS-like TPS enzymes (termites, *Melamspora* fungi), the GGPPS-TPS functional shift is accompanied by an amino acid change in this position.

We then focused on features linked to GGPP cyclization activity. The introduction of a T152I mutation in the neocembrene synthase NtGGPPS6 lead to the formation of the linear diterpene alcohol geranyllinalool from GGPP instead of the cyclic neocembrene. Thus, Thr152 is a core amino acid involved in GGPP cyclization. Our results suggest that after the lysis of GGPP, the arising highly reactive carbocation is stabilized by the hydroxyl group of Thr152 at C-1. This intermediate can then be attacked by the nucleophilic double bond at C-14, followed by proton elimination from C-16. Another explanation of our data is that the mutation of Thr152 results in a difference in the substrate binding position. While substrate positioning in the native protein might promote the nucleophilic attack by the double bond at C-14, yielding a cyclized product, the T152I mutant might instead facilitate the attack of water resulting in linear terpene alcohols. A crystal structure with the substrate or a substrate-like inhibitor is necessary to describe the exact mechanism.

The NtGGPPS6 T152I mutant further let us identify the importance of Ile152 in shifting the ratio of nerolidol enantiomers synthesized from FPP in favor of the *R* enantiomer. The farnesyl isoprenoid backbone is sterically stabilized by isoleucine in an orientation suitable for enantiospecific nucleophilic attack by water at C-3. It appears that isoleucine found at a corresponding position in EneoC contributes to the full enantioselectivity of this enzyme giving rise to pure (3*R*,6*E*)-nerolidol reported from queens [9,57].

Interestingly, EneoE, which also generates neocembrene, lacks Thr152. We suggest that in EneoE, the substrate cyclization proceeds by a different mechanism. The active site is rich in aliphatic amino acids, enabling GGPP pre-folding prior to pyrophosphate cleavage (see Fig 6D). The fact that EneoE produces a completely racemic mixture of nerolidol when supplied with FPP also points to distinct reaction mechanisms in the two enzymes.

With this study, we outline the evolutionary background of terpenoid biosynthesis in termites, and our findings invite further exploration of several related topics. Firstly, to gain mechanistic understanding of the evolutionary transition from IDS to TPS enzymes, confirming our hypothesis based on AlphaFold models using protein structures is required, followed by a thorough mutational analysis of candidate amino acids. Secondly, the search for currently unidentified enzymes which modify the basic terpene skeleton generated by TPSs would elucidate the biosynthesis of the highly complex diterpenes. Thirdly, the observation of general presence of mitochondrial targeting sequence in *E. neotenicus* GGPPS-like proteins and absence in GGPPS (S17 Fig), suggests their possible acquisition during an early duplication event of the GGPPS coding sequence. So far, the subcellular localization of insect IDS-like TPS proteins has not been investigated in any species and represents a thrilling follow-up topic focusing on termite GGPPS-like proteins. Finally, the characterization of the enzyme responsible for producing the queen pheromone (3*R*,6*E*)-nerolidol in *E. neotenicus* queens opens important questions about its regulation and localization at both tissue and subcellular levels, which remain unknown [9].

## 4. Materials and methods

### 4.1. Origin of termites

Colonies of *N. takasagoensis* (Nawa, 1911) and *E. neotenicus* (Holmgren, 1906) were used in this study. *N. takasagoensis* colony was collected at West beach, Ishigaki island, Okinawa Prefecture, Japan (N24°25.886′ E124°04.280′) in September 2021 and kept thereafter at Okinawa Institute of Science and Technology, Japan. Samples for chemical analyses were collected in January 2022 and processed at IOCB Prague.

Colonies of *E. neotenicus* were collected in French Guiana between 2022 and 2024 at two forest localities along the Road to Petit Saut (N5°04.250′ W52°58.770′–N5°04.650′ W53°01.360′) and one locality 13 km south-west of Kourou (N5°06.440′ W52°45.521′).

## 4.2. Chemical analyses

### 4.2.1. Chemical analyses of *N. takasagoensis*.

Entire bodies of 500 *N. takasagoensis* soldiers were extracted overnight in 5 mL of hexane. 1 µL aliquot of soldier extract was analyzed using a TRACE 1310 gas chromatograph coupled to an ISQ LT quadrupole mass spectrometer. The instrument was equipped with a Rxi-5MS column (30 m × 0.25 mm × 0.25 µm). A split/splitless injector was operated in split mode with a 10:1 split ratio, and the injector temperature was set to 250 °C. The carrier gas flow rate was maintained at 1.5 mL × min$^{-1}$. The oven temperature program was as follows: the initial temperature was set to 50 °C and held for 1 min, followed by a ramp of 8 °C × min$^{-1}$ to a final temperature of 320 °C, which was held for 5 min. Mass spectra were acquired at an ionization energy of 70 eV, with a solvent delay of 4 min. The transfer line temperature was set to 260 °C.

### 4.2.2. Chemical analyses of *E. neotenicus*.

In *E. neotenicus*, we sampled 3 neotenic queens, 10 workers, 10 soldiers, and 10 female nymphs (4th stage), and extracted them with 300 µL of hexane for 10 min. One primary king was extracted with 100 µL of hexane for 10 min. A 1 µL aliquot of each extract from individual castes was analyzed using comprehensive gas chromatography coupled with mass spectrometry (GC × GC-MS). The analysis was performed on an Agilent gas chromatograph equipped with a secondary oven, quad-jet thermal modulator, and a Gerstel multipurpose sampler used as an autosampler. A LECO Pegasus BT 4D time-of-flight mass spectrometer was used as the detector. The injection was carried out using a split/splitless injector operated in splitless mode, with the injector temperature set to 250 °C. The chromatographic setup consisted of two columns: in the first dimension, a Rxi-5MS column (30 m × 0.25 mm × 0.25 µm) and in the second dimension, a Rxi-17Sil column (1.5 m × 0.25 mm × 0.25 µm) was used. The oven temperature program was as follows: the initial temperature was set to 50 °C and held for 1 min, followed by a ramp of 8 °C × min$^{-1}$ to a final temperature of 320 °C, which was held for 5 min. The modulation period was set to 3 s, and the transfer line temperature was maintained at 250 °C. A 15 °C offset was used for the secondary oven and a 15 °C offset relative to the secondary oven was used for the modulator.

### 4.2.3. Preparative gas chromatography and NMR.

Unknown polycyclic diterpenes from soldier extracts of *N. takasagoensis* were identified using a combination of preparative gas chromatography (prep-GC) and NMR spectroscopy. The prep-GC system consisted of an Agilent GC connected to a Gerstel preparative fraction collector (PFC) and equipped with a Gerstel program temperature vaporization (PTV) inlet. A 5 µL aliquot of the sample was injected onto a Rxi-5MS column (30 m × 0.53 mm × 1.50 µm). PTV temperature was set to 250 °C for 0.1 min, then ramped at 700 °C × min$^{-1}$ to 320 °C held for 4 min; then the inlet was cooled down at 100 °C × min$^{-1}$ back to 250 °C. Column flow was set to 1.4 mL × min$^{-1}$. After 1.5 min, the inlet pure flow was increased to 50 mL × min$^{-1}$ and held for 3 min. The GC temperature program was set as follows: the starting temperature was set to 200 °C and held for 1 min, then ramped to 320 °C by 15 °C × min$^{-1}$ and held for 3 min. The PFC temperature was set to 320 °C. Fraction collection was performed using coiled glass traps (1 µL capacity) cooled to 0 °C with liquid nitrogen. Each trap was washed with 500 µL of dichloromethane. The collected samples were then evaporated under a stream of nitrogen and re-diluted with deuterated dichloromethane. Six traps were connected to the effluent splitter; however, only three fractions were collected with sufficient purity and quantity for subsequent NMR analysis. NMR spectra were measured on Bruker Avance III HD 600 MHz (600.1 MHz for $^1$H spectra and 150.9 MHz for $^{13}$C spectra) equipped with 1.7 mm TCI MicroCryoProbe. Identification was based on H,H-COSY, H,C-HSQC, H,C-HMBC, and H,H-ROESY 2D spectra.

### 4.2.4. Chiral chromatography.

For the chiral separation of pinene, a 1 µL aliquot of *N. takasagoensis* soldier extract was analyzed using an HP 6850 gas chromatograph equipped with a flame ionization detector (FID) and an HP-Chiral

20β GC column (30 m, 0.25 mm, 0.25 μm) with helium as the carrier gas. The split/splitless injector was operated in splitless mode, and the injector temperature was set to 230 °C. The carrier gas flow rate was maintained at 1.5 mL × min⁻¹. The detector temperature was set to 220 °C, with an $H_2$ flow of 40 mL × min⁻¹ and an air flow of 400 mL × min⁻¹. The temperature program was as follows: an initial temperature of 40 °C held for 20 min, followed by a ramp of 1 °C × min⁻¹ to 100 °C, then a secondary ramp of 30 °C × min⁻¹ to 240 °C, which was held for 10 min. The same setup was used for chiral analysis of limonene, differing only in the temperature program which was set as follows: starting temperature was held for 1 min and then ramped 3 C°/min to 130 °C followed by second temperature ramp 30°C/min to 240 which was held for 10 min. Standards of (+)- and (–)-α-pinene and (+)- and (–)-limonene were used for identification.

For nerolidol analysis, the headspace in the vial from the TPS assay was sampled using a polydimethylsiloxane-coated solid-phase microextraction (SPME) fiber (red fiber, 100 μm; Supelco) for the duration of incubation, followed by desorption in a split/splitless injector operated in splitless mode. The instrument setup for this analysis was the same as for the monoterpene analysis, except for the temperature program: 80 °C held for 1 min, followed by a ramp of 2 °C × min⁻¹ to 200 °C, which was then held for 10 min. For quantitation of nerolidol enantiomers in WT and T152I mutant NtGGPPS6, the peak area of (3R,6E)-nerolidol was divided by sum of peak areas of both enantiomers. The difference in resulting ratios in wild type and mutant enzymes was tested by a two-tailed $t$ test. Each experiment was done in triplicate.

## 4.3. DNA and RNA isolation, library preparation, and sequencing

### 4.3.1. DNA isolation and sequencing of *E. neotenicus*.
High molecular weight DNA for Oxford Nanopore sequencing was isolated by organic extraction from 4 male and 8 female nymphs (whole bodies without digestive tube) of *E. neotenicus*. Deep-frozen tissue samples were homogenized in 600 μL of lysis buffer (10 mM Tris-HCl, 400 mM NaCl, 100 mM EDTA, 1% SDS and 1 mg × mL⁻¹ Proteinase K, pH 8.0) with a PTFE pestle. After overnight incubation at 37 °C, DNA was extracted from aqueous phase by two consecutive extractions with 650 μL phenol:chloroform:isoamyl alcohol (25:24:1) and a reextraction with 600 μL chloroform:isoamyl alcohol (24:1). Phases were separated by centrifugation (5 min, 14,000 $g$, 4 °C). The DNA was then precipitated by addition of NaCl and ethanol to final concentration of 0.2 M and 70%, respectively, incubated for 1 hour at 4 °C, washed with 70% ethanol and resuspended in nuclease-free water. The purity and quantity of isolated DNA were assessed with Nanodrop ND-1000 spectrophotometer and Qubit 4 fluorometer using the dsDNA BR Assay Kit (Thermo Fisher Scientific), the integrity was checked on 0.5% agarose gel in TAE buffer. The samples were used for Oxford Nanopore sequencing analysis on one PromethION flowcell at Novogene UK.

### 4.3.2. RNA isolation and sequencing.
*E. neotenicus* kings, neotenic queens, soldiers, workers, and female nymphs were sampled for RNA isolation. Collected individuals were cold-anesthetized and dissected to remove the digestive tube and, when appropriate, to sample distinct tissues (heads and thorax + abdomen). Samples were stored in RNA later (Thermo Fisher Scientific). Samples were homogenized in TRI Reagent (Sigma-Aldrich) with a PTFE pestle following the manufacturer's protocol. The precipitated RNA was resuspended in DEPC-treated $ddH_2O$. RNA quality and quantity were inspected on Nanodrop ND-1000 UV/VIS spectrophotometer, Qubit 4 fluorometer with the RNA HS Assay Kit (all Thermo Fisher Scientific), and Agilent 2100 Bioanalyzer using the RNA 6000 Nano Kit (Agilent Technologies). The samples were shipped to Novogene UK, where nonstranded poly(A)-selected RNA library preparations and sequencing analysis on Illumina NovaSeq platform with output of 20 million of 2 × 150 bp paired-end reads were conducted.

## 4.4. Bioinformatics and protein structure prediction

### 4.4.1. *E. neotenicus* draft genome assembly and annotation.
The quality of Oxford Nanopore sequencing data was evaluated using fastQC v0.12.1 and NanoPlot v1.30.1, and residual sequencing adapters were removed using Porechop v0.2.4. Draft *E. neotenicus* genome was assembled using Flye v2.9.4 [58] with 5 subsequent polishing iterations, and the estimated genome size was 0.9 Gb, based on an in-house draft genome assembly of a related termite species

*Prorhinotermes simplex* [59]. High-quality Illumina RNA data were mapped to the newly assembled draft genome with Segemehl aligner v0.3.4 [60] and used for an additional round of assembly polishing with Pilon algorithm v1.24 [61]. In a final step, all contigs shorter than 1 kb were removed, and assembly quality was evaluated with Quast v5.0.2 [62], BUSCO v5.4.4 using the insecta_odb10 HMM library [63], and Merqury using Meryl v1.3 [64]. The final draft genome with a total size of 1.38 Gb consisted of 4,512 polished scaffolds with an N50 of 3.2 Mb, and covered 99.4% (97.4% single-copy, 2.0% duplicated, 0.4% fragmented, 0.2% missing) from a total of 1,367 benchmarking universal single-copy orthologs (BUSCO) [63] of insecta_odb10 lineage. The identification of low-complexity genomic regions was performed using the EDTA pipeline v2.2.0 [65] incorporating both RepeatModeler TE prediction and RepeatMasker for genome masking. The masked genome was then used for annotation with BRAKER v3.0.8 gene prediction tool [66] using full RNA-seq data from caste-specific transcriptomic analysis mapped with Segemehl and annotated genes from published isopteran assemblies of *Zootermopsis nevadensis*, *Cryptotermes secundus*, and *Reticulitermes speratus*. Gene annotation provided in total 26,292 gene models covering 92.8% of insect BUSCOs. The genome assembly statistics are summarized in S1 Table.

**4.4.2. Transcriptome assembly.** The *E. neotenicus* transcriptome was assembled *de novo* with RNA-seq data representing heads and body cavities from kings, queens, workers, soldiers, and female nymphs. The data were stripped of artifacts using rCorrector [67] and Trimmomatic v0.32 [68], reads corresponding to ribosomal RNA were filtered out based on mapping to the blacklist from SILVA database [69] using bowtie2 v2.3.4.1 algorithm [70]. The transcriptome was assembled using rnaSPAdes algorithm [71] and the assembly quality was evaluated using rnaQUAST [72] and BUSCO [63]. Open reading frames (ORFs) with at least 100 amino acid residues in size were predicted using Transdecoder (https://github.com/TransDecoder). The SPAdes assembly provided roughly 500,000 unfiltered contigs, covering 98.3% of 1,367 insect BUSCO genes (28.4% single-copy, 69.9% duplicated, and 0.8% fragmented). A similar approach was applied for the *de novo* assembly of *S. halumicus* transcriptome from SRX6715957 data downloaded from NCBI SRA archive and for *de novo* assemblies of transcriptomes of *Spinitermes trispinosus* and *Inquilinitermes inquilinus* from our in-house data deposited at NCBI as SRA within the BioProject PRJNA918850.

**4.4.3. IDS gene discovery.** The prediction of *E. neotenicus* FPPS- and GGPPS-like sequences was carried out in a series of BLAST searches [73] against transcriptome assembly and predicted ORFs using orthologous sequences of three related species (*Z. nevadensis*, *C. secundus*, and *N. takasagoensis*) available in public repositories. The same approach was used for *S. halumicus*, *S. trispinosus*, and *I. inquilinus*. Candidate *E. neotenicus* transcript sequences were used for another round of BLAST searches in BRAKER-annotated gene models. Manual annotation of GGPPS-like sequences was performed by a combination of multiple local alignments in unmasked genome including transcripts that were found in transcriptome assembly but not recognized in predicted gene models, and a detailed inspection of exon junctions in RNA-seq data mapped on unmasked genome using IGV genomic viewer [74].

**4.4.4. *E. neotenicus* expression analysis.** Trimmed RNA-seq data from the heads and thoraces + abdomens of *E. neotenicus* nymphs and neotenic queens were mapped to the annotated genome using STAR v2.7.10b aligner [75]. Read-count matrices were generated with the featureCounts algorithm from rSubread package [76] and normalized according to the TPM (Transcripts Per Kilobase Million) method. Differential gene expression was inspected in pairwise analyses comparing castes and body parts using R packages DESeq2 and edgeR [77,78].

**4.4.5. Transposable elements prediction in the genome of *E. neotenicus*.** The prediction of low-complexity genomic regions and annotation of suspect TE in GGPPS-like loci was performed using a pipeline involving the EDTA first on a genome-wide level, later on a subset of 27 genomic contigs selected based on local alignments of protein-coding sequences of GGPPS-like genes and pseudogenes. Only contigs with full or partial overlaps showing homology over 80% and containing at least 2 consecutive exon-like regions were selected in the panel and used for a focused prediction of transposable elements in the second round of the EDTA pipeline. The genome-scale EDTA annotations were processed in a set of TE enrichment analyses using the BEDtools toolkit and R. Mean TE densities were calculated in both, all protein-coding and GGPPS-like only gene regions expanded by 50 kb from each side as well as in whole-genome separated

ndto 100 kb windows in a set of 1,000 random permutations. The 100 kb window was then separated into strand-oriented 1 kb bins to inspect the TE distribution relative to gene centers followed by TE family representation in GGPPS-like loci compared to whole-genome [79].

**4.4.6. Phylogenetic analyses and prediction tools.** Publicly available sequences of FPPS and GGPPS homologs were retrieved from NCBI. Online blastp and tblastn searches of nr (nonredundant protein sequences) and TSA databases limited to Blattodea were run with *Z. nevadensis* FPPS and GGPPS amino acid sequences as queries (XP_021942300, XP_021933767). The same searches were also run with *Z. nevadensis* decaprenyl-diphosphate synthase subunit 1 as query (XP_021928794). Local blastp and tblastn searches were used for searching *R. speratus* datasets deposited at figshare [28], our in-house datasets from *E. neotenicus*, *S. trispinosus*, and *I. inquilinus*, and the in-house assembled transcriptome from *S. halumicus*. Sequence logos for FARM and SARM motifs in FPPS- and FPPS-like proteins were generated in R using the ggseqlogo package [80]. For phylogenetic analyses, amino acid sequences with a minimum length of 150 amino acids and comprising both FARM and SARM motifs were selected. Unless otherwise stated, sequence alignment was performed using Clustal Omega [81] run from SeaView (v4.7) [82], and the resulting alignments were inspected and trimmed manually in AliView [83]. Phylogenetic trees were reconstructed with IQ-TREE 2 [84] using the ModelFinder option and setting the bootstrap replicate number to 1,000 unless stated otherwise.

We used MEME [85] (v5.5.7) to detect IGMs in a dataset comprising functionally characterized insect GGPPSs, several single-copy GGPPSs from a diverse selection of insect lineages, and Blattodean sequences from the ancestral GGPPS clade. Regular expressions were generated for the motifs and their boundaries were set as defined for IPP-binding motifs (IBM) by Rebholz and colleagues [34]. The presence or absence of GGPPS and GGPPS-like sequences in individual termite species was determined based on a phylogeny for sequences ≥ 150 amino acids that included FARM and SARM motifs. In species with shorter sequences and/or not comprising FARM and SARM motifs, we used the motifs previously identified as characteristic for insect GGPPSs to distinguish between GGPPS and GGPPS-like sequences.

Putative MTSs were predicted with MitoFates [86] in newly reported genes from *E. neotenicus*. In addition, we checked the completeness of annotated ORFs in publicly available *N. takasagoensis* GGPPS and GGPPS-like mRNA sequences. In two cases (NtGGPPS1B, NtGGPPS3B), we applied MTS predictions to putative proteins coded by ORFs starting with upstream ATG codons not considered in the published annotation. In two other cases (NtGGPPS2B, NtGGPPS4), the ORF was found to be positioned at the very beginning of the transcript. Therefore, it was impossible to assess the completeness of these ORFs and to consider MTS prediction relevant for these proteins.

All modeled GGPPS sequences were trimmed based on alignment with *Homo sapiens* GGPPS sequence (UniProt accession O95749) and were modeled as a single subunit with three $Mg^{2+}$ ions, although it can be assumed that *in vivo*, the enzymes exist as dimers or hexamers [87]. Structure models for EneoC, EneoE, and NtGGPPS6 with corresponding substrates, and the ancestral sequence at node 2 were generated using Alphafold 3.0 provided by e-INFRA project. The GGPP substrate was included in models of NtGGPPS6, EneoE, and the node 2, the FPP substrate in case of EneoC. NtGGPPS7 and EneoJ were modeled by homology-based modeling using modeler 10.6 software [88] with the structure of *Homo sapiens* GGPPS (PDB Entry 2Q80) as a template. The active site cavity was identified and highlighted using KVfinder-web [89] implementing parKVFinder v 1.2.0 [90]. We probed whole structures with probe size in 1.4 Å and probe size out 4 Å. Structures were aligned and processed using UCSF ChimeraX 1.6.1 [91]. For comparison of FPPS- and FPPS-like active sites, the crystal structure 1YV5 (PDB ID: pdb_00001yv5) of human FPPS was downloaded from the RCSB Protein Data Bank. Termite FPPS- and FPPS-like proteins were modeled with three $Mg^{2+}$ ions using AlphaFold 3. The crystal structure, together with the two models, was aligned and processed using UCSF ChimeraX 1.6.1. We then displayed all interactions between $Mg^{2+}$ ions and the surrounding amino acids within 2.7 Å.

## 4.5. Quantitative proteomics

Heads and thoraces + abdomens (with digestive tubes removed) from four nymphs and four neotenic queens of *E. neotenicus* were subjected to label-free quantitative (LFQ) mass spectrometry (MS) analysis. Each sample was homogenized

in 500 μL of homogenization buffer consisting of 100 mM triethylammonium bicarbonate buffer (TEAB; Sigma-Aldrich) and 2% sodium deoxycholate (SDC; Sigma-Aldrich) in a 1.5-mL Eppendorf tube using FastPrep-24 homogenizer (MP Biomedicals). The homogenate was centrifuged to remove cell debris and the supernatant was used for analysis. Further sample processing and MS analysis were performed as previously described [92]. Briefly, protein concentration was determined using a BCA protein assay kit (Thermo Fisher Scientific). Cysteines were reduced with Tris(2-carboxyethyl) phosphine hydrochloride and blocked with methyl methanethiosulfonate. The samples were digested with porcine trypsin. Then, the samples were acidified with trifluoroacetic acid to a final concentration of 1%. SDC was removed by extraction with ethylacetate and subsequently with hexane. Peptides were desalted using a Michrom C18 column. Dried peptides were resuspended in 25 μL of water containing 2% acetonitrile and 0.1% trifluoroacetic acid. Peptides were separated by nano-liquid chromatography on a Dionex Ultimate 3000 system and analyzed on an Orbitrap Fusion Tribrid mass spectrometer (both Thermo Fisher Scientific).

The data were evaluated with MaxQuant version 2.2.0.0 using LFQ algorithms [93,94] and the Andromeda search engine [95]. The key criteria used in the data analysis were a false discovery rate (FDR) of 0.01 for proteins and peptides, a minimum length of seven amino acids, a fixed modification (methylthio), and variable modifications of N-terminal protein acetylation and methionine oxidation. The data were locally evaluated against selected databases of nonredundant protein sequences that were downloaded from NCBI on February 13, 2023. The in-house database of *E. neotenicus* protein sequences was based on protein models generated by BRAKER (see *E. neotenicus* draft genome assembly and annotation) complemented by a set of manually annotated IDS genes. The local database is available along with the raw data and result files in the PRIDE repository of the ProteomeXchange Consortium (see Data availability). The data were processed in Perseus version 2.0.7.0 [96]. Results representing standard contaminants, reverse sequences (decoys), and sequences identified only with modified peptides were discarded. The dataset containing proteins with at least two valid LFQ values in at least one experimental group was further analyzed. The dataset was $\log_2$ transformed. Missing LFQ values were replaced from the normal distribution (width 0.3; downshift 1.8) and histograms before and after the data imputation were examined. A heatmap with hierarchical clustering was used to check the uniformity of the sample proteomes between the two subgroups. Significant differences were determined by a permutation-based two-sided *t* test with an error-corrected *p*-value (FDR = 0.05; S0 = 0.1). The number of randomizations in the analysis was 1,000.

### 4.6. Heterologous expression

Coding sequences of candidate TPS and IDS proteins were codon-optimized for expression in *S. cerevisiae*, synthesized, and cloned into MCS-1 of pESC-URA by Genscript. The resulting plasmids coded for recombinant proteins with N-terminal FLAG-tag and were used for expression in *S. cerevisiae*. For expression of recombinant proteins with N-terminal hexahistidine tag in *E. coli*, the coding sequences were recloned in MCS-1 of pRSFDuet-1 (Novagen). Proteins in which mitochondrial targeting peptides were predicted by MitoFates were expressed in the presumed cleaved form. Amino acid sequences of all expressed proteins are listed in S14 Table.

**4.6.1. Expression in *S. cerevisiae* and metabolite detection.** The JWY501 strain of *S. cerevisiae* was used for all yeast experiments [97]. Prior to transformation, the JWY501 strain was grown in a complete medium (YPD) containing 20 g × L⁻¹ Tryptone, 10 g × L⁻¹ Yeast Extract (both from Duchefa Biochemie), and 20 g × L⁻¹ glucose. Selective media (SCE) used for cultivation of transformed JWY501 contained 1.92 g × L⁻¹ Yeast Synthetic Drop-out Medium Supplements without Uracil, 6.7 g × L⁻¹ Yeast Nitrogen Base without amino acids (both from Sigma-Aldrich), and 20 g × L⁻¹ glucose. Selective media used for expression had the same composition as SCE, except that 20 g × L⁻¹ galactose was used instead of glucose to induce expression of the gene of interest.

Yeast transformations were performed using the lithium acetate PEG method [98]. Shortly, cells were pre-cultured overnight in YPD medium at 30 °C with shaking at 200 rpm. The following day, the overnight culture was diluted to an optical density (OD600) of 0.1 in fresh YPD medium and grown at 30 °C with shaking at 200 rpm until reaching an OD600 of ~0.5. Cells were harvested by centrifugation at 3,000 *g* for 5 min, washed twice in 25 mL of sterile $H_2O$, and pelleted again under

the same conditions. The resulting cell pellet was resuspended in 1 mL of sterile water, centrifuged at 13,000 $g$ for 30 s, and resuspended in 1 mL of sterile water. For each transformation, 100 µL of resuspended cells were mixed with 240 µL of 50% (w/v) PEG 3350, 36 µL of 1 M LiAc, 100 ng of plasmid DNA, and 100 µg of denatured salmon sperm DNA (all from Sigma-Aldrich) resulting in a final volume of 460 µL. The transformation mixture was gently mixed, incubated at 42 °C for 40 min, and centrifuged at 13,000 $g$ for 30 s. Cells were washed with sterile water, plated onto a selective agar medium, and incubated at 30 °C for 3–5 days to allow for colony formation.

Cells carrying the plasmid of interest were pre-cultured in a selective medium (YNB - URA + 2% glucose) for 1 day at 30 °C with shaking at 200 rpm. The following day, the culture was diluted to an initial optical density (OD600) of 0.3 in fresh selective medium and grown at 30 °C with shaking until the OD600 reached ~1. At this point, cells were harvested by centrifugation, resuspended in YNB - URA + 2% galactose, and incubated to induce expression of the gene of interest. Cultures were sampled and extracted with hexane either on day 1 after induction (1 mL sample + 1 mL hexane) or on day 7 after induction (100 µL sample + 1 mL hexane). The organic phase was analyzed by GC × GC-MS.

A 1 µL aliquot of hexane yeast extract was analyzed using an Agilent comprehensive gas chromatograph equipped with a secondary oven and a quad-jet thermal modulator, coupled with a LECO Pegasus 4D Time-of-Flight (TOF) mass spectrometer. The injection was performed using a split/splitless injector in splitless mode, with the injector temperature set to 250 °C. A 10 °C offset was used for the secondary oven, and a 10 °C offset relative to the secondary oven was used for the modulator. The primary column used was ZB-5MS (30 m × 0.25 mm × 0.25 µm), and the secondary column was Rxi-17Sil (1.980 m × 0.25 mm × 0.25 µm). Helium was used as the carrier gas, with a flow rate of 1 mL × min⁻¹. The temperature program started at 50 °C, held for 1 min, then ramped at 8 °C × min⁻¹ to 320 °C and was held for 5 min. The transfer line temperature was set to 260 °C. The electron ionization ion source operated at 70 eV, with the ion source temperature maintained at 220 °C. A solvent delay of 400 s was applied.

**4.6.2. Expression in *E. coli* and protein purification.** For protein purification, pRSFDuet-1 plasmids were used for transformation of *E. coli* Lemo21(DE3). Bacteria were grown in 2.5 L Tunair baffled shake flasks (IBI Scientific) filled with 0.4 L auto-induction medium, composed of 15 g × L⁻¹ yeast extract, 10 g × L⁻¹ peptone, 10 g × L⁻¹ glycerol, 2 g × L⁻¹ D-lactose, 0.5 g × L⁻¹ D-glucose, 100 mM KH₂PO₄, pH 7.5, 50 mM NaCl, 25 mM (NH₄)₂SO₄ and 3 mM MgSO₄. The culture (starting OD600 ~ 0.01, total culture volume 2–2.4 L) was supplemented with 30 mg × L⁻¹ kanamycin, 30 mg × L⁻¹ chloramphenicol, and 0.1 mM L-rhamnose, and a drop of Antifoam A concentrate (Merck) was added to each flask. Cultures were shaken for 6 hours at 37 °C, 250 RPM (capped with aluminum foil lids), and further for 16–20 hours at 17 °C (capped with cellulose paper tissues). The cells were harvested by centrifugation and stored at −80 °C.

After thawing, the cells were suspended with 10 volumes of lysis buffer (100 g × L⁻¹ glycerol, 0.3 M NaCl, 50 mM KH₂PO₄, pH 6.7, 10 mM imidazole, 2 mM MgSO₄) using a tissue homogenizer. The suspension was supplemented with 0.5 mg × mL⁻¹ chicken egg lysozyme, 2,000 KU × L⁻¹ DNase I (Merck), and 1 mM PMSF and incubated while rolling. After 1 hour, 10 mM β-mercaptoethanol (ME) was added and incubation while rolling continued for another 2 hours at 4 °C. Cells were broken by four passages through Avestin Emulsiflex C3 and the resulting crude lysate was clarified by consecutive centrifugations (30,000 $g$, 5 and 10 min). The supernatant was filtered through 0.6 µm pore glass filter and pump-loaded at 5 mL × min⁻¹ onto HisTrap Fast Flow 5 mL column (Cytiva) using ÄKTA start or ÄKTA pure chromatograph (Cytiva). The column was subsequently washed with 250 mL of lysis buffer with 5 mM ME and the protein was eluted by a four-step gradient of 10% elution buffer (EB; i.e., lysis buffer with 5 mM ME and 500 mM imidazole), 20% EB, 30% EB, and 100% EB. Protein concentration was estimated using a calculated extinction coefficient at 280 nm.

## 4.7. TPS and IDS *in vitro* assays

For enzyme activity assays, the reaction mixture in a 2 mL glass vial consisted of 50 µM substrate and roughly 5 µM recombinant protein in 10 mM 3-(N-morpholino)propanesulfonic acid (MOPS), 5 mM MgCl₂, 1 mM dithiothreitol, 10% glycerol, pH 7.0, in a final volume of 100 µL. The substrates used in TPS assays were GPP, (*E,E*)-FPP, or (*E,E,E*)-GGPP

 

(Merck or Cayman Chemicals). For IDS assays, the substrates included a mixture of 50 μM IPP with 50 μM DMAPP, GPP, or (*E*,*E*)-FPP (Merck or Cayman Chemicals). The reactions were incubated at 30 °C for 1 hour. In TPS assays, the head-space in the vial was sampled by a polydimethylsiloxane-coated SPME fiber for the duration of incubation (red fiber, 100 μm; Supelco).

The fiber containing the sample was desorbed and analyzed using a TRACE 1310 gas chromatograph coupled to an ISQ LT quadrupole mass spectrometer. An SPME liner was used in the inlet, with the desorption temperature set to 250°C. The fiber was desorbed in the inlet for 5 min to ensure complete sample desorption. The GC and MS methods were identical to those described previously.

Isoprenoid pyrophosphates were analyzed using a Vanquish Horizon HPLC system coupled with an Orbitrap ID-X mass spectrometer. The separation was performed on an ACCUCORE VANQUISH C18+ column (1.5 μm, 150 × 2.1 mm). As mobile phase A, 10 mM ammonium bicarbonate with 0.1% ammonium hydroxide was used. Acetonitrile with 0.1% ammonium hydroxide was used as mobile phase B. The column flow rate was set to 0.35 mL × min$^{-1}$. The system was equilibrated with 95% mobile phase A and 5% mobile phase B before the chromatographic run. The gradient was pro-grammed as follows: 5% B for the first 5 min, ramp to 100% B over 5.5 min, 100% B held for 7.3 min. The column tem-perature was maintained at 40 °C throughout the analysis. Electrospray ionization in negative mode was used with an ionization voltage of −3,000 V applied from 0.2 min. The gas settings were as follows: Sheath gas, 50 Arb; Auxiliary gas, 10 Arb; Sweep gas, 1 Arb. Additional parameters included an ion transfer tube temperature of 325 °C and a vaporizer temperature of 350 °C.

## 4.8. Selection analysis and ancestral sequence reconstruction

To investigate the role of selection pressures in the evolution of terpene synthase genes in termites from canonical GGPPS progenitors, we utilized the packages HyPhy (v2.5.29) and PAML (v4.10.7) to perform selection analysis [99,100]. Full-length GGPPS and GGPPS-like coding sequences from termites were selected from both terpene-releasing Neoisop-tera and early branching termite clades without terpenoid production. To generate a tree for a selection analysis and ancestral sequence reconstruction, translated coding sequences were first aligned using ClustalOmega (v1.2.3) under default parameters and trimmed using Trimal (v1.5) with a gap threshold of 0.5. The alignment was used to infer a maximum-likelihood protein phylogeny with IQTree2 (UFboot x1000) using the substitution model JTT + F + I + G4 selected by ModelFinder and the singleton GGPPS of the cockroach *Blatella germanica* as an outgroup. To generate a codon alignment, nucleotide coding sequences were aligned using ClustalOmega (v1.2.3) implemented in the Translation Align function of Geneious software. To test for clade-wide differences in the selection regimes acting on canonical GGPPS and the expanded GGPPS-like gene family, an analysis was conducted using HyPhy RELAX [101]. Termite canonical GGPPS clades were designated as the reference branches and the GGPPS-like clade containing TPSs characterized in this study as the test branches. dN/dS ratios (ω) of specific branches were estimated and changes in selection between test and reference branches were tested by fitting a selection intensity parameter (k) to the data. Statistical evidence for a change in selection intensity in the test branches was then detected by comparison of the likelihood of the fitted k to that of a null model in which no change in selection intensity has occurred (k = 1). To test for evidence of episodic diversifying selection in the evolution of the termite GGPPS-like gene family, an analysis was conducted using HyPhy aBSREL [102] implemented in the HyPhy package testing all branches in the GGPPS-like clade. Branches predicted to be under positive diversifying selection using HyPhy aBSREL were again tested for positive selection using branch-site tests implemented using the CODEML program from PAML [100], where the likelihoods of three-ratio foreground models (allowing positive selection on a proportion of sites on the foreground branches) were tested against the likelihood of tree-wide two-ratio null models (no positive selection). The BEB method was also employed in branch-site tests to predict codons under positive selection pressure on different foreground branches. Significance for all selection analysis was assessed using Holm-Bonferroni corrected *p*-values with a significance threshold of 0.05.

Ancestral termite TPS sequences were reconstructed in PAML, using the above phylogeny and codon alignment as input. A stretch of 7 amino acids (LLNQTDI) was found to be present in all ancestral sequences across the tree, obviously reflecting an EneoO-specific insertion which did not occur in any other extant sequence. Therefore, we omitted these amino acids from ancestral sequences which are presented in S12 Table. Accordingly, the ancestral sequence B used for modeling the protein structure lacks this insertion.

## Supporting information

**S1 Fig. Representative GC×GC chromatograms of body washes from different _E. neotenicus_ castes.** The chromatograms show the retention window where terpenoids were detected. 1, β-farnesene; 2, α-farnesene; 3, (6E)-nerolidol; 4, unidentified diterpene; 5, β-springene; 6, (E,E,E)-neocembrene, 7, unidentified diterpene. The chromatogram visualizes the responses to diagnostic ions for mono-, sesqui-, and diterpenes ($m/z = 68 + 69 + 93 + 204 + 272$). Mass spectra of the identified terpenes are shown below in S2 Fig. Complete chromatographic data are available from OSF (https://osf.io/rkdy9).
(TIF)

**S2 Fig. Mass spectra of terpenes identified in the body washes of different _E. neotenicus_ castes.** Compound numbering refers to that in Figs 1 and S1. For compound 7 (unidentified diterpene), we could not obtain a reliable mass spectrum.
(TIF)

**S3 Fig. Representative GC chromatogram of body washes from _N. takasagoensis_ soldiers and workers.** The chromatogram visualizes the responses to diagnostic ions for mono-, sesqui-, and diterpenes ($m/z = 68 + 69 + 93 + 204 + 272$). Mass spectra and NMR spectra of the identified terpenes are shown below in S4–S8 Figs. Complete chromatographic data are available from OSF (https://osf.io/rkdy9).
(TIF)

**S4 Fig. Identification of compound a from _N. takasagoensis_ soldiers as (+)-α-pinene.** A. Chiral chromatogram comparing the retention of compound a with standards of (−) and (+)-α-pinene. GC-FID was equipped with cyclodextrin chiral column (HP CHIRAL β, 30 m, 0.25 mm, 0.25 μm, 19091G-B233E). One μL of sample was injected in splitless mode following this temperature program: 40 °C held for 20 min, ramped 1 °C/min to 100 °C, ramp at 30 °C/min to 240 °C, held for 20 min. B. Mass spectrum of compound a. Complete chromatographic data are available from OSF (https://osf.io/rkdy9).
(TIF)

**S5 Fig. Identification of compound a from _N. takasagoensis_ soldiers as (+)-limonene.** A. Chiral chromatogram comparing the retention of compound b with standards of (−) and (+)-limonene. GC-FID was equipped with cyclodextrin chiral column (HP CHIRAL β, 30 m, 0.25 mm, 0.25 μm, 19091G-B233E). One μL of sample was injected in splitless mode following this temperature program: 40 °C held for 1 min, ramped 3 °C/min to 130 °C, ramp at 30 °C/min to 240 °C, held for 10 min. **B.** Mass spectrum of compound b. Complete chromatographic data are available from OSF (https://osf.io/rkdy9).
(TIF)

**S6 Fig. Identification of compound c from N. takasagoensis soldiers as 14α-acetoxy-3β,20 β -dipropionoxy-kampa-6(7),8(9)-diene. A.** Mass spectrum. **B.** Key 2D NMR correlations and assigned $^1$H and $^{13}$C NMR spectra. **C.** $^1$H NMR spectra. **D.** APT NMR spectrum. **E.** HMBC NMR spectra. **F.** HSQC NMR spectrum. **G.** COSY NMR spectrum. **H.** ROESY NMR spectrum. Complete NMR data are available from OSF (https://osf.io/rkdy9).
(TIF)

**S7 Fig. Identification of compound d from _N. takasagoensis_ soldiers as 3α,9β,13α-tripropionoxy-trinervita-11(12),15(17)-diene. A.** Mass spectrum. **B.** Key 2D NMR correlations and assigned $^1$H and $^{13}$C NMR spectra.

**C.** $^1$H NMR spectra. **D.** APT NMR spectra. **E.** HMBC spectra. **F.** HSQC NMR spectra. **G.** COSY spectra. **H.** ROESY NMR spectra. Complete NMR data are available from OSF (https://osf.io/rkdy9).
(TIF)

**S8 Fig. Identification of compound e from *N. takasagoensis* soldiers as 3α,9β,13α-tripropionoxy-11α(12β)-epoxy-trinervita-15(17)-ene. A.** Mass spectrum. **B.** Key 2D NMR correlations and assigned $^1$H and $^{13}$C NMR spectra. **C.** $^1$H NMR spectra. **D.** APT NMR spectrum. **E.** HMBC NMR spectra. **F.** HSQC NMR spectrum. **G.** COSY NMR spectra. **H.** ROESY NMR spectrum. Complete NMR data are available from OSF (https://osf.io/rkdy9).
(TIF)

**S9 Fig. Phylogeny and function of FPPS homologs in termites. A.** Phylogenetic tree inferred from amino acid sequences of FPPS homologs identified in Blattodea (five termite and two cockroach species) and two other polyneopteran orders, Orthoptera and Phasmatodea. The maximum-likelihood tree was reconstructed with IQ-TREE 2 using JTT + G4 model; bootstrap values (500 replicates) greater than 50 are shown as node support. Arrow marks the FPPS duplication predating the diversification of Blattodea. Asterisks highlight the two paralogous sequences EneoA and EneoB from *Embiratermes neotenicus* studied here with respect to their function. Accession numbers are provided in S2 Table. **B.** FARM and SARM motif sequence logos in FPPS- and FPPS-like proteins, as inferred from 197 amino acid sequences of Blattodea listed in S3 Table. **C.** A crystal structure of human FPPS active site and a model of the active site in EneoB and EneoA proteins showing coordinated $Mg^{2+}$ ions. **D.** HPLC chromatogram demonstrating the IDS activity of EneoB. The purified enzyme was incubated with isopentenyl pyrophosphate (IPP) and dimethylallyl pyrophosphate (DMAPP), IPP and geranyl pyrophosphate (GPP), or IPP and farnesyl pyrophosphate (FPP). Only the substrate combinations giving rise to any prenyl pyrophosphate products are shown. The chromatograms visualize the selected m/z 245.00, 313.06, 381.12, and 449.19. No TPS activity was recorded in EneoB and no TPS or IDS activity was observed in EneoA. Complete chromatographic data and phylogenetic tree file are available from OSF (https://osf.io/rkdy9).
(TIF)

**S10 Fig. Alignment of IDS catalytic motifs in FPPS- and FPPS-like amino acid sequences.** Motif boundaries and sequence regular expressions are displayed as defined by Rebholz and colleagues (2023) for insect IDS sequences. Residues in consensus with the regular expression are shaded blue. Termite species are displayed in bold text. IBM, IPP-binding motif. FARM, first aspartate-rich motif. SARM, second aspartate-rich motif.
(TIF)

**S11 Fig. Genomic landscape surrounding the *E. neotenicus* GGPPS-like genes and pseudogenes.** The figure shows gene organization of full or partial genomic contigs (boldfaced, starting with prefix ENC and contig size in brackets) represented by horizontal line with position coordinates at both sides. Exons are numbered and shown as black (genes), gray (pseudogenes), or red boxes (chimeric duplicates), regions corresponding to genes and pseudogenes are marked with black or gray arrows while regions masked by RepeatMasker at genome-wide scale are shown as pink boxes. Sequences homologous to CACTA TIR TE_00000081 DNA transposon predicted with the EDTA pipeline are marked with green arrows. Blue scale bar represents 5 kb.
(TIF)

**S12 Fig. Maximum-likelihood phylogenetic trees inferred from alignments of GGPPS amino acid sequences. A.** Taxon sampling focusing on Neoisoptera-specific GGPPS multiplication. The tree was reconstructed using IQ-TREE with the JTT + F + R4 substitution model; bootstrap values (1,000 replicates) greater than 50 are shown as node support. The tree was rooted with a dragonfly GGPPS sequence (*Ischnura elegans*). Roman numerals denote the paraphyletic GGPPS clade I and monophyletic GGPPS-like clades II–IV. See Supplementary Table S6 for source sequences. The tree file is available from OSF. **B.** Origin of GGPPS duplications in Neoisoptera and the butterfly genus *Heliconius.* A

maximum-likelihood phylogenetic tree was inferred from an alignment of amino acid sequences using IQ-TREE with the JTT + I + G4 substitution model; bootstrap values (1,000 replicates) greater than 50 are shown as node support. The tree was rooted with a dragonfly GGPPS sequence (*Ischnura elegans*). See S6 Table for source sequences. The tree file is available from OSF (https://osf.io/rkdy9).
(TIF)

**S13 Fig. Analysis of transposable element (TE) enrichment in GGPPS-like loci based on whole-genome TE prediction using the EDTA pipeline. A.** Mean TE frequency in 100 kb windows surrounding GGPPS-like paralogs (orange line), FPPS-like paralogs (pink line), all protein-coding genes (blue line) and genomic background represented by a histogram of TE frequencies in 100 kb genomic regions generated within 1,000 random permutations (gray bars). **B.** Mean TE density according to distance from the gene center (dashed vertical line) in GGPPS-like loci (orange) and protein-coding genes (blue). TE frequencies were calculated in strand-oriented 1,000 bp bins, satellite peaks in GGPPS-like loci correspond to proximal paralogs located in the same genomic locus with a mean distance of 23 kb from the gene. **C**. TE family representation in 100 kb windows surrounding the GGPPS-like loci (orange) compared to the abundance on whole-genome level (blue). Graph bars represent the relative frequencies of TE families among all predicted TEs. **D.** Sequence conservation of the CACTA TIR DNA transposable element linked with GGPS-like genes. Sequences corresponding to CACTA TIR TE_00000081 predicted with RepeatModeler (green) and homologous sequences found in the proximity of GGPPS-like genes and pseudogenes in *E. neotenicus* (yellow) are shown as arrows, conserved regions are represented by dark color. Red triangles above sequence diagrams represent indels relative to the CACTA TIR TE_00000081 reference. Orthologs from related species *N. takasagoensis* (Ntak) and *R. speratus* (Rspe) found by BLAST search against the complete NCBI nucleotide collection restricted to Blattodea are included. **E**. The sequence of CACTA TIR TE_00000081 predicted with RepeatModeler, latin numbers I–III are used to specify three regions with various levels of conservation. Custom code used for A–C is available from OSF (https://osf.io/rkdy9).
(TIF)

**S14 Fig. HPLC chromatograms demonstrating the IDS activity in canonical GGPPSs EneoJ (yellow) and NtGGPPS7 (green).** The purified enzymes were incubated with IPP and DMAPP, IPP and GPP, or IPP and FPP. The chromatograms visualize the selected m/z 245.00, 313.06, 381.12, and 449.19. The products resulting from IDS activity are underlined with black lines. Complete chromatographic data are available from OSF (https://osf.io/rkdy9).
(TIF)

**S15 Fig. Expression of IDS and IDS-like genes in *E. neotenicus* queens and nymphs at the level of transcript (A) and protein (B). A.** Differential expression analysis comparing transcript abundances of IDS and IDS-like genes in heads or thoraces and abdomens of neotenic queens and queen-destined 4th stage female nymphs. Mean $\log_2$ TPM values represent three biological replicates, colors of connecting lines correspond to $\log_2$ fold changes between the two castes. Only genes which showed significant regulation between the two castes in DESEq analysis are presented (padj < 0.05). Expression data for all IDS/IDS-like genes are listed in S8 Table. **B.** Quantitative proteomic analysis of IDS/IDS-like proteins in thoraces and abdomens of queen-destined 4th stage female nymphs and neotenic queens. Mean $\log_2$ intensity values represent four biological replicates. Colors correspond to $\log_2$ fold changes between the two castes. Only proteins which conform to the permutation-based FDR 0.05 are shown; points with no connecting lines represent proteins which were only detected in samples from queen thoraces and abdomens. Results of proteomic analysis for all IDS/IDS-like proteins from all tissues are listed in S9 Table.
(TIF)

**S16 Fig. Screening of enzymatic activity of GGPPS-like enzymes from *Nasutitermes takasagoensis*.**
Sequences were expressed in *S. cerevisiae* strain JWY501. 1 mL of culture was extracted with 1 mL of hexane and

1 μL aliquot of hexane extract was measured by comprehensive gas chromatography mass spectrometry. No specific products were detected in any of the strains apart from neocembrene (a) in the NtGGPPS6-expressing strain. **A.** NtGGPPS1A- NtGGPPS3B. **B.** NtGGPPS4-NtGGPPS6. Complete chromatographic data are available from OSF (https://osf.io/rkdy9).
(TIF)

**S17 Fig. Alignment of IDS catalytic motifs in GGPPS and GGPPS-like protein sequences.** Boundaries of motifs are displayed as defined by Rebholz and colleagues [34]. Insect GGPPS-specific regular expression for each motif was generated based on the canonical insect GGPPS sequences displayed in the alignment using MEME (v5.5.7). Residues in consensus with the regular expression are shaded blue. Termite species are printed in bold text, amino acid residues for confirmed TPS proteins are set in bold. MTS, presence or absence of a mitochondrial targeting peptide as predicted by MitoFates (v1.2). IGM, insect GGPPS motif. FARM, first aspartate-rich motif. SARM, second aspartate-rich motif.
(TIF)

**S1 Table. General statistics of *E. neotenicus* draft genome assembly and in silico annotated gene models.**
(XLSX)

**S2 Table. Source of FPPS- and FPPS-like sequences identified in Blattodea and polyneopteran outgroups used for phylogenetic tree reconstruction.**
(XLSX)

**S3 Table. Source of Blattodea FPPS- and FPPS-like sequences used to generate the FARM and SARM sequence logos.**
(XLSX)

**S4 Table. Source of GGPPS and GGPPS-like nucleotide sequences in selected termites and nontermite hexapod outgroups.**
(XLSX)

**S5 Table. *Stylotermes halumicus* GGPPS and GGPPS-like nucleotide sequences assembled *de novo* from the SRA data deposited as SRX6715957.**
(XLSX)

**S6 Table. Source of GGPPS and GGPPS-like sequences used for amino acid phylogeny.**
(XLSX)

**S7 Table. Source of GGPPS and GGPPS-like sequences identified in 56 species of termites.**
(XLSX)

**S8 Table. Differential expression analysis of IDS and IDS-like genes in female nymphs and queens of *E. neotenicus*.**
(XLSX)

**S9 Table. Quantitative proteomic analysis of IDS and IDS-like proteins in female nymphs and queens of *E. neotenicus*.**
(XLSX)

**S10 Table. Source of GGPPS and GGPPS-like sequences used for the selection analysis.**
(XLSX)

**S11 Table. RELAX analysis of selection intensity in GGPPS and GGPPS-like clades.**
(XLSX)

**S12 Table. Reconstructed ancestral amino acid sequences at nodes 1, 2, and 3.**
(XLSX)

**S13 Table. Proportion of *R* and *S* enantiomers of (6*E*)-nerolidol upon incubation of wt NtGGPPS6 and T152I mutant with FPP.**
(XLSX)

**S14 Table. Amino acid sequences of recombinant proteins from *E. neotenicus* (Eneo) and *N. takasagoensis* (Nt).**
(XLSX)

## Acknowledgments

We would like to thank the Keasling lab (Joint BioEnergy Institute, University of California, Berkeley) for providing the *Saccharomyces cerevisiae* strain JWY501. We are grateful to Esra Kaymak (Okinawa Institute of Science and Technology, Japan) for assistance with samples of *Nasutitermes takasagoensis*, and to former IOCB interns Nanyun Zhang and Guillame Masson for participation in experiments.

## Author contributions

**Conceptualization:** Natan Horáček, Dorothea Tholl, Robert Hanus, Jitka Štáfková.

**Data curation:** Natan Horáček, Ondřej Lukšan, Zarley Rebholz, Karel Harant, Radek Pohl, Jitka Štáfková.

**Formal analysis:** Natan Horáček, Ondřej Lukšan, Zarley Rebholz, Karel Harant, Radek Pohl, Jitka Štáfková.

**Funding acquisition:** Robert Hanus.

**Investigation:** Natan Horáček, Ondřej Lukšan, Karel Harant, Radek Pohl, Lana Mutabdžija-Nedelcheva, Daniel Jungwirth, Jan Křivánek, Anna Amirianová, Pavlína Kyjaková, Jitka Štáfková.

**Methodology:** Natan Horáček, Ondřej Lukšan, Jitka Štáfková.

**Project administration:** Jitka Štáfková.

**Resources:** Simon Hellemans, Jan Křivánek, Thomas Bourguignon.

**Supervision:** Dorothea Tholl.

**Validation:** Natan Horáček.

**Visualization:** Natan Horáček, Robert Hanus.

**Writing – original draft:** Natan Horáček, Ondřej Lukšan, Zarley Rebholz, Robert Hanus, Jitka Štáfková.

**Writing – review & editing:** Natan Horáček, Dorothea Tholl, Robert Hanus, Jitka Štáfková.

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
