## [Editor Report · Decision Letter 0]

27 Jan 2026

Dear Dr Štáfková,

Thank you for submitting your manuscript entitled "Evolutionary origin of terpenoid biosynthesis in termites" for consideration as a Research Article by PLOS Biology.

Your manuscript has now been evaluated by the PLOS Biology editorial staff, as well as by an academic editor with relevant expertise, and I'm writing to let you know that we would like to send your submission out for external peer review.

Once your full submission is complete, your paper will undergo a series of checks in preparation for peer review. After your manuscript has passed the checks it will be sent out for review. To provide the metadata for your submission, please Login to Editorial Manager (https://www.editorialmanager.com/pbiology) within two working days, i.e. by Jan 29 2026 11:59PM.

Kind regards,

Roli Roberts

Roland Roberts, PhD

Senior Editor

PLOS Biology

rroberts@plos.org

---

## [Decision Letter · Decision Letter 1]

11 Mar 2026

Dear Dr Štáfková,

Thank you for your patience while your manuscript "Evolutionary origin of terpenoid biosynthesis in termites" was peer-reviewed at PLOS Biology. Please note that I am currently handling your manuscript since my colleague Roland Roberts is out of the office this week. I am very sorry for the delays that you have experienced during the peer review process. Your manuscript has now been evaluated by the PLOS Biology editors, an Academic Editor with relevant expertise, and by four independent reviewers.

In light of the reviews, which you will find at the end of this email, we would like to invite you to revise the work to thoroughly address the reviewers' reports.

As you will see, the reviewers are positive about your study and agree that it is interesting and well done. Reviewer #1 is mostly satisfied with the manuscript as-is, although would like to see additional reporting details for the NMR data. Reviewer’s #2 and #3 provide a list of suggestions that are textual in nature, such as toning down several overstatements, adding reporting details and adding additional clarifications/discussions. Reviewer #4 is the most critical reviewer and raises concerns with the strength of the evolutionary data and its interpretation. After discussions with the Academic Editor, we will not make the request in major point 1 to include functional characterization of further genes essential for the revision, but we ask that a more cautious interpretation is provided. Regarding major points 2 and 3, we agree that the requested analyses would be beneficial additions to the paper. Therefore, we ask that additional data is included to test whether transposable elements are enriched in the regions of GGPPS-like genes, as well as including a phylogenetic tree of other insect GGPPS genes, especially those involved in terpene synthesis.

Given the extent of revision needed, we cannot make a decision about publication until we have seen the revised manuscript and your response to the reviewers' comments. Your revised manuscript is likely to be sent for further evaluation by all or a subset of the reviewers.

We expect to receive your revised manuscript within 2 months. Please email us (plosbiology@plos.org) if you have any questions or concerns, or would like to request an extension.

**IMPORTANT - SUBMITTING YOUR REVISION**

*Re-submission Checklist*

*Published Peer Review*

*PLOS Data Policy*

*Blot and Gel Data Policy*

Sincerely,

Richard

Richard Hodge, PhD

rhodge@plos.org

On behalf of:

Roland Roberts, PhD

rroberts@plos.org

REVIEWS:

Reviewer #1: I really like this manuscript!

The authors address the interesting and challenging subject of terpene/terpenoid biosynthesis in termites. This study sits within the fascinating area of terpene biosynthesis in insects, which shows significant differences from terpene biosynthesis in plants and microorganisms, and is in many ways more diverse and rich.

The manuscript uses the Introduction section to provide a scholarly summary of terpene biosynthesis in insects, which sets up the current study very well. Complementary expertise is applied to the subject, including organic chemistry, biochemistry and evolutionary biology to produce a very comprehensive and rigorous piece of work. Fascinatingly, it would seem that TPSs in termites have arisen from GGPPSs, whether they are used to produce sesquiterpenoids (i.e. nerolidol) or diterpenes (cembrene). This contrasts nicely with FPPS-derived diterpene synthases from other insects.

Although I have ticked the minor corrections box, I have no corrections for the main manuscript itself, which I judge to be acceptable for publications as is (I rarely select this option!). I did notice that in the supplementary section, Fig S6 C and D are labelled the wrong way round. Also, for the APT NMR data, it would be helpful to state the pos/neg phasing of the C types (CH/CH3 vs C/CH2), as this is the opposite phasing is often employed. Additionally, on the topic of NMR, it is stated that identifications are based on numerous multidimensional NMR, but these data are not shown. For completeness, it would be good to see these spectra.

Reviewer #2: In the presented manuscript Horacek et al. describe the identification, biochemical characterization, and possible evolutionary origin of terpene biosynthesis in Neoisoptera. The authors start by chemically characterizing different specimens from the termite species Embiratermes neotenicus and Nasutitermes takasagoensis. They find the previously identified queen pheromone nerolidol end the diterpene neocembrene in the extracts from E neotenicus queens and two monoterpenes as well as three oxidized diterpenoids in the soldiers of N. takasagoensis. Next, they investigated the genomes and transcriptomes of termites for candidate terpene synthase genes. This analysis showed a neoisoptera-specific duplication of geranylgeranyldiphosphate synthases (GGPPS), which, within neoisoptera, shows great consecutive diversification. In the genome assembly of E neotenicus, which was generated for this study, these duplicated GGPPS-like genes occur close to transposable elements, which are likely responsible for the numerous duplications of this gene family. Two GGPPS-like genes, which were relatively overexpressed in queens of E neotenicus with respect to nymphs of that species, were functionally characterized. Indeed, these two enzymes showed enantioselective nerolidol production, and neocembrene production, respectively. The authors also characterize six GGPPS-like genes obtained from N. takasagoensis soldier transcriptomes, of which only one was active upon heterologous expression. However, the produced enzyme was a selective neocembrene synthase, which presents a plausible precursor for the identified soldier diterpenoids. The authors additionally conduct a selection analysis, comparing selection pressures on GGPPS-like terpene synthase genes to the ones influencing canonical GGPPSs. This analysis shows a great decrease in negative selection comparing the former to the latter, pointing to a decrease in purifying selection, allowing higher mutation rates and thus functional evolution. Finally, the authors identify a functional switch in the characterized N. takasagoensis terpene synthase, which abolishes the selective neocembrene production activity and results in the production of geranyllinalool, a linear product.

This study is very interesting and of importance in the field of specialized metabolism. It is very comprehensive and goes all the way from chemical and genomic analysis to characterizing terpene synthase enzymes in vitro, shedding light on their evolutionary history within termites and the genetic mechanisms on diversification. The article is clearly written and mostly easy to follow despite the described work being highly multidisciplinary. The overwhelming majority of the conclusions made in the manuscript are robustly backed by data, which is comprehensively presented in the supplementary information.

Thus, in conclusion, I recommend publication of this article after addressing some minor concerns and suggestions, which are listed below.

General comments:

Usually the identifiers for partial figures (e.g., A, B, etc) are at the top left corner of each element. Throughout this manuscript they are at the bottom left which, at least for me, is very confusing to look at. Please consider moving the identifiers to the top-left corners.

Details:

-Abstract: "Termites produce the most diverse array of terpenoids among metazoans, comprising over 200 structures." - This is an incorrect statement. There are thousands of structurally diverse terpenoids from sponges and corals.

-Abstract: "and the precursor of polycyclic defensive diterpenes (E,E,E)-neocembrene in Nasutitermes takasagoensis" this has not been proven yet, please soften the language to something like "plausible precursor".

-Page 4 lines 63-66: In these two sentences the authors state that utilizing classical TPS genes originating from horizontal gene transfer is an alternative to utilizing classical TPS genes. I think this statement should be removed, as receiving a gene via horizontal gene transfer at an unclear point in time seems like an arbitrary distinction versus having carried in a genome for a longer (unspecified) time. Many genes have been transferred via HGT at some point in time, possibly too long ago to detect it conclusively.

-Page 4, lines 83-86: Here the authors state that neocembrene is the precursor to polycyclic, more complicated diterpenoids. To my knowledge, there is no direct evidence for this (e.g., feeding studies with labeled neocembrene showing incorporation into the diterpenoids, or enzyme characterization by in viro assays or knockouts). So, this should be phrased more as a hypothesis. If there is indeed direct evidence, please cite this here specifically.

-Page 7 lines 147-149: The authors conclude from the phylogenetic analysis and the biochemical inactivity that the tandem duplicated FPPS genes are not relevant in terpene biosynthesis. I don't think this can be definitively stated here. An ancestral tandem duplication preserved in all of Blattodea could point to some function, otherwise this gene might have been lost at least in some species. Apparent biochemical inactivity might result from a variety of reasons (improper protein folding, missing PTMs, incorrect substrates tested) and even if the gene from the tested species has lost its function this does not automatically mean that this find is true for all termites. Please soften the language here.

-Figure 4: The colors for negative control and one of the chiral standards are too similar, it took me a while to make sense of the chromatograms. Consider a different, lighter shade of gray for the S standard. Both R and S could be the same color, as they are also specifically labeled. Please explicitly state in the figure caption that the absolute configuration for neocembrene was not determined in this study. As it is now the figure alone will suggest it is R in both cases as it is drawn that way.

-Figure 6: Please also label T152I and WT traces for the assays of NtGGPPS6 and GGPP.

-Page 16 lines 419-422: The authors describe mitochondrial targeting sequences and refer to figure F13. However, I cannot identify any reference to mitochondrial targeting sequences in this figure. Out of curiosity: Do the authors think that the GGPPS-like enzymes are indeed transported to the mitochondrion?

-Page18 lines 463-469: While the functional switch by the single amino acid variant T152I is interesting, the presented data does not allow for a detailed analysis of the specific interactions of the amino acid sidechains in threonine or isoleucine. A very likely scenario that is not discussed here is that the initial substrate binding position is impacted in a way that either promotes attack by water, leading to linear terpene alcohols or by the double bond at C-14, leading to cyclization. Different binding modes can also explain the shift from almost racemic to strongly enantioenriched water attack in the production of nerolidol. On the other hand, it is quite unlikely that the saturated carbon side chain of isoleucine contributes significantly to the electronic stabilization of a carbocationic intermediate. Hence, I would encourage the authors to also briefly discuss steric effects on the observed enzymatic activities.

Page 27 lines 699-700: What software was used for substrate docking?

Supplementary Information:

Please provide more information on the structure elucidation of the diterpenoids of N. takasagoensis. Currently, there is only 1D NMR data presented and a list of signals mapped to the structures. Please show the 2D-NMR spectra and specify the correlations that allowed the unambiguous structure elucidation (a figure with the typical single- and double head arrows should suffice).

Reviewer #3: This manuscript provides an interesting and detailed description of terpenoid biosynthesis and its evolutionary origins in termites. The text was well-written and the figures (except for some minor issues described below) were effective for the information presented. The data presented appear to provide good support for the conclusions drawn.

Main Text:

Line 73: Reference 20 describes TPS activity from a GPPS, not an FPPS as indicated in the text. See also: 10.1073/pnas.0503277102.

Line 221: This is the first mention of EDTA as a software. Perhaps spell out: Extensive de novo TE Annotator (EDTA).

Line 241: Which other substrates?

Line 292: Incorporate HyPhy RELAX into the previous sentence, i.e., "...with HyPhy RELAX. Our analysis showed..."

Line 550: alpha symbol missing

Line 877-879: The annotated genome would be utilized by a much wider audience if it was submitted to a public data repository such as NCBI rather than "hidden" in the ASEP data repository that only readers of your published manuscript are likely to discover. I.e., As is, this data will not become incorporated into NCBI's BLAST databases. I strongly encourage the authors to submit the annotated genome to NCBI.

Supporting Figures issues:

Fig. S6A, S7A, and S8A have issues will displaying m/z labels, and the lines themselves.

Fig. S6C and D appear to be reversed relative to their legend information.

Fig. S15 the termites are cut off at the top.

Supporting Tables issues:

Table S1, the BUSCO scores don't indicate which parameter they are referring to.

Table S6, is labeled Table S5 at the top of the worksheet.

Reviewer #4: The manuscript titled "Evolutionary origin of terpenoid biosynthesis in termites" examines the genetic and molecular basis of terpenoid biosynthesis in two termite species. Termites are known to produce more than 200 types of terpenoids; however, the biosynthetic pathways and the enzymes involved in their production have not been well elucidated.

The authors focused on two termite species, Embiratermes neotenicus and Nasutitermes takasagoensis, and analyzed the terpenoid compounds produced in different castes of each species. They combined genome sequencing with tissue- and caste-specific transcriptome analyses to identify candidate genes involved in terpene biosynthesis. In subsequent phylogenetic analyses and functional assays of selected candidate genes, the authors found that some members of a clade of GGPPS-like genes possess TPS activity and are capable of synthesizing terpenes detected in the studied termite species.

Furthermore, the authors report that the GGPPS-like clade experienced extensive gene duplication and that these genes tend to harbor higher densities of TEs in their genomic regions. Additional evolutionary analyses suggest that genes in the GGPPS-like clade tend to experience more relaxed purifying selection and that the branch leading to this clade shows evidence of positive selection. Based on these findings, the authors conclude that gene duplications, potentially associated with TE enrichment, followed by subsequent diversification enabled termites to produce a diverse array of terpenes.

Overall, the manuscript is well written and I enjoyed reading it. Knowledge of terpene biosynthesis in insects has been accumulating in recent years, and this study provides an important contribution toward understanding the genetic and molecular basis of terpene biosynthesis in termites. While I am convinced that the identified genes indeed possess TPS activity in termites, I have several major concerns regarding the evolutionary interpretation of the results, as outlined below.

Major comments

1. The contribution of GGPPS-like gene duplication to terpene diversification remains unclear.

One of the central claims of this manuscript is that the duplication of the GGPPS-like clade, followed by diversifying selection, led to the diversification of terpene synthases in termites. The authors indeed demonstrate that several members of the GGPPS-like gene family exhibit TPS activity, and evolutionary analyses detect diversifying selection acting on this clade.

However, the genes that were functionally characterized in this study represent only a subset of the entire GGPPS-like clade, even within the two focal species examined here. Moreover, among the tested genes, a substantial proportion did not exhibit TPS activity. Therefore, while the current results clearly demonstrate that some GGPPS-like genes have acquired TPS activity, they do not yet fully support the broader conclusion that the expansion of the GGPPS-like clade itself underlies the remarkable diversity of terpene biosynthesis in termites. Additional functional characterization of GGPPS-like paralogs or a more cautious interpretation of the evolutionary implications would strengthen the manuscript.

2. The evidence supporting the involvement of TEs in GGPPS-like gene duplication is insufficient.

The authors suggest that the expansion of the GGPPS-like clade may have been mediated by TEs because TE sequences are frequently found near GGPPS-like loci. However, the analyses presented do not clearly demonstrate that TE density around these loci is enriched relative to the genome-wide background distribution. Although the manuscript mentions a comparison with FPPS loci, the supporting data are not shown, making it difficult to evaluate whether the observed pattern is statistically meaningful.

Furthermore, the Discussion suggests that the presence of MTS in GGPPS-like proteins supports the TE-mediated duplication hypothesis. However, the observed distribution of MTS could also arise if the ancestral gene of the GGPPS-like clade acquired an MTS after a gene duplication event, regardless of whether the duplication was mediated by TE activity. Therefore, the presence of MTS alone does not appear to provide strong support for the proposed TE-mediated duplication scenario. A more direct comparison of TE density with genome-wide expectations or analyses of duplication structures would be necessary to substantiate this hypothesis.

3. The evolutionary position of GGPPS-like genes is not sufficiently discussed in the broader context of insect TPS evolution.

In this study, the evolutionary analyses of GGPPS and GGPPS-like genes are largely restricted to termite lineages. However, terpene synthases derived from IDS have been reported in multiple insect lineages. Therefore, it would be informative to place the termite GGPPS-like genes within a broader phylogenetic context that includes TPS and IDS genes from other insect taxa.

Such a comparison would clarify whether the GGPPS-like genes represent a lineage-specific innovation in termites or whether they are related to TPS genes previously identified in other insects. Including a broader phylogenetic framework would help strengthen the evolutionary interpretation and highlight the novelty of the findings.

Minor comments

Line 607: Genome assemblies are usually polished using genomic resequencing reads rather than RNA-seq data. Polishing a genome with RNA-seq reads may be problematic because transcript reads originate from expressed genes only and can bias correction toward highly expressed transcripts. In addition, this approach may potentially collapse sequence differences among closely related gene copies or paralogs, which is particularly concerning in the case of expanded gene families such as the GGPPS-like genes analyzed in this study.

EneoE is still expressed at a certain level in nymphs, although almost no nerolidol or neocembrene was detected in this caste (Fig. 1A and C). Could EneoE be involved in another pathway that synthesizes other terpene-related compounds?

What do the asterisks in Fig. 2D represent?

---

## [Editor Report · Decision Letter 2]

31 Mar 2026

Dear Dr Štáfková,

Thank you for your patience while we considered your revised manuscript "Evolutionary origin of terpenoid biosynthesis in termites" for publication as a Research Article at PLOS Biology. This revised version of your manuscript has been evaluated by the PLOS Biology editors and the Academic Editor.

Based on our Academic Editor's assessment of your revision, we are likely to accept this manuscript for publication, provided you satisfactorily address the following data and other policy-related requests.

IMPORTANT - please attend to the following:

a) Please make your Title more informative and declarative, with an active verb. We suggest: "Expansion of the geranylgeranyl pyrophosphate synthase gene family underlies the evolution of terpenoid biosynthesis in termites"

b) Please address my Data Policy requests below; specifically, we need you to supply the numerical values underlying Figs 1AB, 2B (treefile), 4ABC, 5A (treefile), 6B, S1, S2, S3, S4AB, S5AB, S6ACD, S7ACDEFGH, S8ACDEFGH, S9A (treefile), S9D, S12 (treefiles), S13ABC, S14, S15AB, S16, either as a supplementary data file or as a permanent DOI’d deposition.

c) Please cite the location of the data clearly in all relevant main and supplementary Figure legends, e.g. “The data underlying this Figure can be found in S1 Data” or “The data underlying this Figure can be found in https://zenodo.org/records/XXXXXXXX" (also note that, where the Fig panel displays chromatography and NMR data, you can simply cite your OSF deposition).

d) Please make any custom code available, either as a supplementary file or as part of your data deposition.

e) Please include the URLs of your funders in the Financial Disclosure statement.

We expect to receive your revised manuscript within two weeks.

*Published Peer Review History*

*Press*

Sincerely,

Roli Roberts

Roland Roberts, PhD

Senior Editor

rroberts@plos.org

PLOS Biology

DATA POLICY:

Regardless of the method selected, please ensure that you provide the individual numerical values that underlie the summary data displayed in the following figure panels as they are essential for readers to assess your analysis and to reproduce it: Figs 1AB, 2B (treefile), 4ABC, 5A (treefile), 6B, S1, S2, S3, S4AB, S5AB, S6ACD, S7ACDEFGH, S8ACDEFGH, S9A (treefile), S9D, S12 (treefiles), S13ABC, S14, S15AB, S16. NOTE: the numerical data provided should include all replicates AND the way in which the plotted mean and errors were derived (it should not present only the mean/average values).

CODE POLICY

Per journal policy, if you have generated any custom code during the course of this investigation, please make it available without restrictions. Please ensure that the code is sufficiently well documented and reusable, and that your Data Statement in the Editorial Manager submission system accurately describes where your code can be found. More information on our Code Policy, what and how to share can be found here: https://journals.plos.org/plosbiology/s/code-availability

DATA NOT SHOWN?

---

## [Editor Report · Decision Letter 3]

9 Apr 2026

Dear Dr Štáfková,

Thank you for the submission of your revised Research Article "Expansion of the geranylgeranyl pyrophosphate synthase gene family underlies the evolution of terpenoid biosynthesis in termites" for publication in PLOS Biology. On behalf of my colleagues and the Academic Editor, Chris Jiggins, I'm pleased to say that we can in principle accept your manuscript for publication, provided you address any remaining formatting and reporting issues. These will be detailed in an email you should receive within 2-3 business days from our colleagues in the journal operations team; no action is required from you until then. Please note that we will not be able to formally accept your manuscript and schedule it for publication until you have completed any requested changes.

Sincerely,

Roli Roberts

Senior Editor

PLOS Biology

rroberts@plos.org